# Dual stem cell therapy synergistically improves cardiac function and vascular regeneration following myocardial infarction

Soon-Jung Park [1,9], Ri Youn Kim[2,9], Bong-Woo Park [3,4,9], Sunghun Lee [2,9], Seong Woo Choi[1], Jae-Hyun Park[4], Jong Jin Choi[1], Seok-Won Kim[5], Jinah Jang[6], Dong-Woo Cho[5], Hyung-Min Chung[1], Sung-Hwan Moon[1,7], Kiwon Ban [2] & Hun-Jun Park[3,4,8]

Since both myocardium and vasculature in the heart are excessively damaged following myocardial infarction (MI), therapeutic strategies for treating MI hearts should concurrently target both so as to achieve true cardiac repair. Here we demonstrate a concomitant method that exploits the advantages of cardiomyocytes derived from human induced pluripotent stem cells (hiPSC-CMs) and human mesenchymal stem cell-loaded patch (hMSC-PA) to amplify cardiac repair in a rat MI model. Epicardially implanted hMSC-PA provide a complimentary microenvironment which enhances vascular regeneration through prolonged secretion of paracrine factors, but more importantly it significantly improves the retention and engraftment of intramyocardially injected hiPSC-CMs which ultimately restore the cardiac function. Notably, the majority of injected hiPSC-CMs display adult CMs like morphology suggesting that the secretomic milieu of hMSC-PA constitutes pleiotropic effects in vivo. We provide compelling evidence that this dual approach can be a promising means to enhance cardiac repair on MI hearts.

[1] Department of Medicine, Konkuk University School of Medicine, Seoul 05029, Republic of Korea. [2] Department of Biomedical Sciences, City University of Hong Kong, Tat Chee Avenue, Kowloon, Hong Kong SAR. [3] Department of Medical Life Science, College of Medicine, The Catholic University of Korea, 222 Banpo-daero, Seocho-gu, Seoul 137701, Republic of Korea. [4] Division of Cardiology, Department of Internal Medicine, Seoul St. Mary's Hospital, The Catholic University of Korea, 222 Banpo-daero, Seocho-gu, Seoul 137701, Republic of Korea. [5] Department of Mechanical Engineering, Pohang University of Science and Technology, 77 Cheongam-ro, Hyogok-dong, Nam-gu, Pohang 37673, Republic of Korea. [6] Department of Creative IT Engineering and School of Interdisciplinary Bioscience and Bioengineering, Pohang University of Science and Technology, 77 Cheongam-ro, Hyogok-dong, Nam-gu, Pohang 37673, Republic of Korea. [7] Research Institute, T&R Biofab Co. Ltd, 237, Siheung, Republic of Korea. [8] Cell Death Disease Research Center, College of Medicine, The Catholic University of Korea, 222 Banpo-daero, Seocho-gu, Seoul 137701, Republic of Korea. [9] These authors contributed equally: Soon-Jung Park, Ri Youn Kim, Bong-Woo Park, Sunghun Lee. Correspondence and requests for materials should be addressed to S.-H.M. (email: sunghwanmoon@kku.ac.kr) or to K.B. (email: kiwonban@cityu.edu.hk) or to H.-J.P. (email: cardioman@catholic.ac.kr)

Myocardial infarction (MI) is a fatal disorder that inflicts a permanent loss of cardiomyocytes (CMs) and scar tissue formation, resulting in irreversible damage to cardiac function ensued by heart failure[1]. While cardiac regeneration is considered unfeasible with current medical options, accumulating evidence in animal models as well as clinical trials continue to demonstrate that stem cells could offer new opportunities for treating MI hearts[2].

Among them, human mesenchymal stem cells (hMSCs) have long been considered a promising candidate for cell-based therapy owed to their beneficial paracrine factors such as vascular endothelial growth factor (VEGF), fibroblast growth factor 2 (FGF2), and hepatocyte growth factor (HGF) that promote angiogenesis, neovascularization, and cell survival[3]. hMSCs are also known to secrete potent anti-fibrotic factors including matrix metalloproteinases 2, 9, and 14, which inhibit the proliferation of cardiac fibroblasts thereby attenuating fibrosis[4].

In tandem, CMs derived from human pluripotent stem cells (hPSC-CMs), which include both human embryonic stem cells (hESCs) and human induced pluripotent stem cells (hiPSC), are propitious due to their similarities with human primary CMs apposite to expressions of cardiac-specific genes, structural proteins, and ion channels as, well as spontaneous contraction[5,6]. Several preclinical studies have shown that hPSC-CMs successfully engraft, align, and couple with host myocardium in a synchronized manner to improve cardiac function[6–8].

Since the heart is an organ composed of cardiac muscles and blood vessels, both cardiac muscles and vasculatures in the heart were excessively damaged following MI. Thus therapeutic strategies for treating MI hearts should be focused to comprehensively repair all that together for achieving true cardiac repair[2]. The principality behind cell-based cardiac regeneration therapy should adhere to the same principles as well.

Hence, in this study, we develop a multipronged approach aiming to concurrently rejuvenate both the myocardium and vasculatures utilizing both hiPSC-CMs and hMSCs. We hypothesize that while intramyocardially injected hiPSC-CMs would restore heart function by engraftment with the host myocardium, epicardially implanted hMSC patches (hMSC-PA) would simultaneously enhance vascular regeneration through consistent secretion of angiogenic paracrine factors in MI-induced hearts. We demonstrate that the dual approach of applying hiPSC-CMs as well as hMSC-PA lead to a significant improvement of cardiac function and enhancement of vessel formation post MI. Of note, not only do the implanted hMSC-PA significantly increase the retention of intramyocardially injected hiPSC-CMs but also preserve injured host CMs via paracrine release of cytokines and growth factors, resulting in functional improvement. Surprisingly, the majority of injected hPSC-CMs neighboring hMSC-PA display the rectangular-shaped morphology of adult-like CM and significant expression of Gja1, a major gap junction protein, suggesting that hMSC-PA promotes CM maturation in vivo. Furthermore, we identify that paracrine factors released from hMSC-PA constitute the key contributors with pleiotropic effects, including pro-angiogenesis, anti-inflammation, anti-fibrosis, and CM maturation. These results carry significant implications for stem cell therapy in cardiac repair, highlighting the need for intricate and strategic designs to achieve complete restoration.

## Results

**Selection of optimal hMSCs and their characterization**. To select the best candidate of hMSCs in terms of secretion of paracrine factors, we examined distinct types of hMSCs isolated from different sources such as human turbinate, human adipose tissue, and human bone marrow and compared the concentration of VEGF secreted to their conditioned medium. As a result, we found that concentration of VEGF released from hMSCs derived from human bone marrows were substantially higher than other types of hMSCs (Supplementary Fig. 1A). Subsequently, we characterized hMSCs from bone marrows and found that bone marrow-derived hMSCs exhibited evident phenotypes of hMSCs. Cultured bone marrow-derived hMSCs displayed a homogeneous spindle-like shape, which is a typical cell morphology of hMSCs and express abundant expression of several specific markers for hMSCs, such as CD73, CD105, CD90, and CD44 (Supplementary Fig. 1B).

**hMSC-PA secrets paracrine factor**. First, to examine the viability of hMSCs within the hMSC-PA, we generated hMSC-PA (Supplementary Fig. 2A) and performed the live/dead staining. As a result, we verified that that the majority of hMSCs within the hMSC-PA remained alive and less than one-tenth of cells were dead, indicating that hMSC-PA can maintain the viability of hMSCs. Next, to confirm whether hMSC-PA could efficiently release paracrine factors from embedded hMSCs, we cultured the hMSC-PA in vitro, collected supernatants, and performed VEGF ELISA assay (Supplementary Fig. 2B). As a result, we verified that hMSC-PA released VEGF over time. We observed that the concentration of VEGF released from hMSC-PA increased in a time-dependent manner, suggesting that hMSC-PA is capable of releasing sufficient amounts of cytokines (Supplementary Fig. 2B).

**hMSC-PA increases the expressions of multiple factors**. To explore the paracrine effects of hMSC-PA in MI hearts in vivo, we first performed gene expression analyses with rat heart tissues harvested 7 days after hMSC-PA implantation into MI hearts (Fig. 1a–d). qRT-PCR results show that hMSC-PA significantly upregulated the expression of several angiogenesis-related genes, such as vascular endothelial growth factor A (VEGFa), insulin-like growth factor 1 (IGF-1), fibroblast growth factor 2 (FGF2), placental growth factor (PLGF), angiopoietin 1 (Ang1), and angiopoietin 2 (Ang2), and cluster of differentiation 31 (CD31) in comparison with the MI control hearts, as well as the cell-free patch implanted group (Fig. 1e). In addition, while expression levels of pro-inflammatory-related genes such as interferon gamma (IFNG), interleukin 1 beta (IL-1b), and tumor necrosis factor alpha (TNFα) were not significantly altered, the expression levels of anti-inflammatory-related genes including transforming growth factor beta 1 (TGFβ1) and interleukin 10 (IL10) were significantly increased in hMSC-PA implanted hearts (Fig. 1f). Of interest, hMSC-PA significantly downregulated the expression of several anti-fibrosis-related gene including collagen type 1 (COL1) and collagen type 3 (COL3) and increased the expression of tissue inhibitors of metalloproteinase 2 (TIMP-2) (Fig. 1g). These qRT-PCR data clearly suggest that hMSCs-PA could induce significant paracrine effects for multiple factors regarding angiogenesis, inflammation, and fibrosis in MI hearts.

**hMSC-PA is only effective in improving angiogenesis**. Given the instant increased expression of several angiogenesis-releated genes, we sought to investigate whether hMSC-PA could improve vascular regeneration in MI hearts. To determine the effects of hMSC-PA, we perfused isolectin B4 (IsB4) conjugated with green fluorescent dye into the heart tissues to visualize the vessels prior to tissue harvest at 8 weeks. Consistent with the qRT-PCR results, assessment of fluorescent images showed that the number of capillaries in the infarct zones of hMSC-PA implanted hearts were significantly higher than the untreated control group suggesting hMSC-PA have significant effects on vascular regeneration in MI hearts (Fig. 2a). To ensure that the illustrated

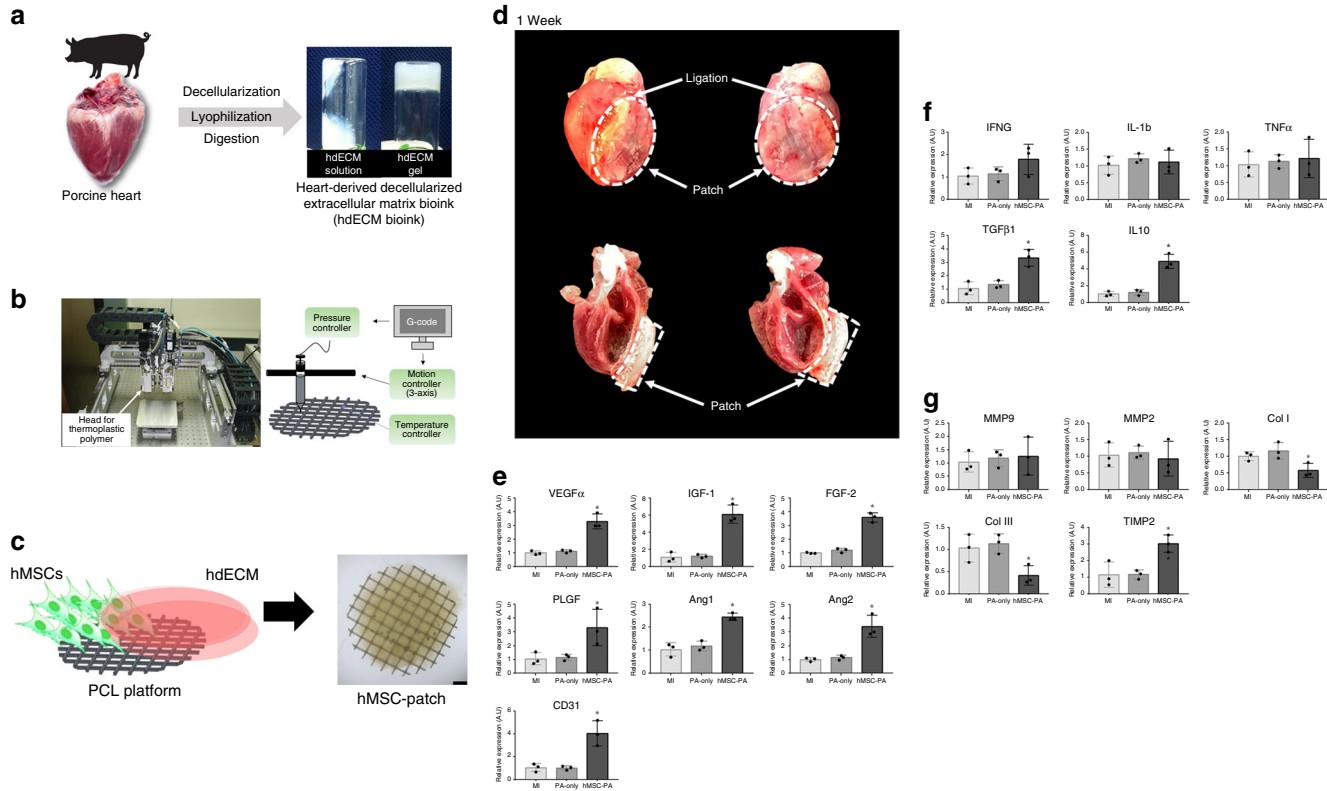

**Fig. 1** Implantation of hMSC-PA into MI hearts enhances the expression of multiple factors. **a** Preparation of heart-derived decellularized extracellular matrix (hdECM) bioink. **b** Macroscopic view and illustration of 3D printing system used to produce PCL platform. **c** Schematic illustration of human mesenchymal stem cell patch (hMSC-PA). Scale bar: 4 cm. **d** Epicardially implanted hMSC-PA in MI heart at 1 week. **e-g** Quantitative real-time polymerase chain reaction analysis of relative mRNA expression of multiple factors in the myocardium at 1 week after hMSC-PA implantation in MI induced hearts. **e** angiogenesis (**f**) inflammation, (**g**) fibrosis. The y-axis represents relative mRNA expression of target genes to GAPDH. A.U. indicates arbitrary units. The data are represented as mean ± SEM. $^+ p < 0.05$ compared with MI group, $^\# p < 0.05$ compared with PA-only group; $n = 3$ biologically independent samples per group. One-way ANOVA was used for statistical analyses. Control: MI control, PA-only: cell-free hdECM patch, and hMSC-PA: hMSC-loaded patch

vasculogenic were exclusively attributed by hMSC-PA, we performed a TUNEL assay using rat heart tissues harvested 8 weeks post implantation. The results revealed that the majority of hMSCs were located within the patch and remained alive (Supplementary Figs 3, 4). hMSCs were active as less than one-tenth of the cells exhibited TUNEL signal (Supplementary Fig. 4). These results suggest that the observed angiogenic effects were attributed by of hMSC-PA, which was viable for at least 8 weeks following implantation.

Next, to evaluate cardiac function and cardiac remodeling, we performed echocardiography on a regular basis. While there was a substantial increase of capillary density in hMSC-PA implanted MI hearts, the echocardiography results demonstrated that hMSC-PA did not induce functional improvement in MI hearts. Both ejection fraction (hMSC-PA: 33.93% ± 1.66% vs. 32.21% ± 0.67% in control group, $n = 7$ animals; $p < 0.05$; one-way ANOVA was used for statistical analyses) and fractional shortening (hMSC-PA: 14.07% ± 0.81% vs. 11.56% ± 0.94% in control group, $n = 7$ animals; $p < 0.05$; one-way ANOVA was used for statistical analyses.) were not significantly different with the untreated control group (Fig. 2b). Based on these results, we postulated that although implantation of hMSC-PA is very effective for vascular regeneration, it is insufficient to improve cardiac function and requires additional cell sources to further enhance heart function.

**Generation of cardiomyocyte derived from hiPSCs.** Owing to several previous studies reporting the promising effects for

restoring cardiac function in MI hearts, we sought to test hiPSC-CMs as an additional cell type to treat MI hearts. Accordingly, we generated hiPSC-CMs by using a small-molecule-based 2D differentiation method (Supplementary Fig. 5)[9–11]. Spontaneous beating was observed around differentiation day 7 (Supplementary Fig. 5B and Supplementary Movies 1, 2). Immunostaining demonstrated that most differentiated hiPSC-CMs displayed CM-specific marker proteins, such as TNNT2 and ACTN2 (Supplementary Fig. 5C). Furthermore, quantification analyses performed via flow-cytometry analysis demonstrated an upwards of 98% of hiPSC-CMs were positive for TNNT2, and ~84% of hPSC-CMs expressed MYL2, a well-known marker for ventricular CMs suggesting successful generation of highly purified hiPSC-CMs (Supplementary Fig. 5D).

**Dual approach increases cardiac function and angiogenesis.** Subsequently, we investigated the therapeutic potential of the dual approach by intramyocardially injecting hPSC-CMs and implanting hMSC-PA (Supplementary Movie 3). After the induction of MI by LAD ligation, we generated five experimental groups receiving: (i) Sham control, (ii) untreated control, (iii) hiPSC-CM intramyocardial injections, (iv) hMSC-PA implantations, or (v) both hiPSC-CMs and hMSC-PA, and compared their therapeutic effects. First, echocardiography results showed that cardiac function in the combined treatment group was significantly higher than hiPSC-CM only or hMSC-PA-only group, as determined by ejection fraction (hiPSC-CMs (CM): 35.95% ± 1.85% vs. combined (CM + PA): 43.15% ± 0.52% vs.

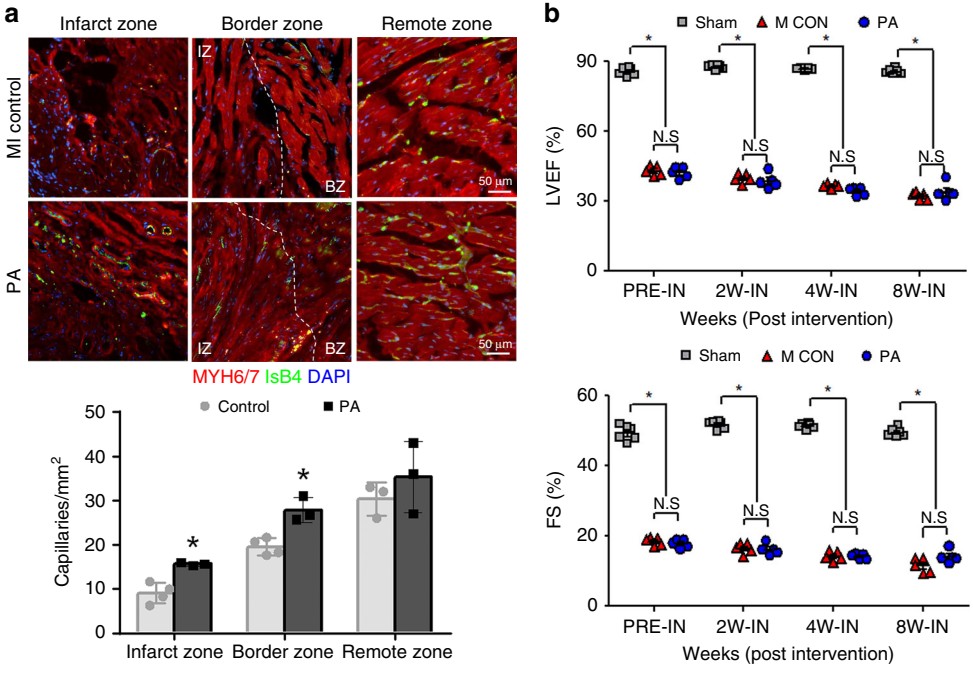

**Fig. 2** hMSC-PA implantation only improved vascular regeneration. **a** hMSC-PA implanted to MI hearts were perfused with GFP-conjugated IsB4 to visualize the vessels, IsB4 (green). Representative images of capillaries on the infarct zone, border zone, and the remote zone at 8 weeks after MI and their quantification summary. For quantification, the number of capillaries on five randomly selected fields in each heart was counted. The number of capillaries was counted per mm² within the infarct zone. Scale bars: 50 μm. The data are represented as mean ± SEM. *$p < 0.05$ compared with control group; $n = 5$ biologically independent samples per group. $T$ test was used for statistical analyses. **b** Rats undergoing MI were implanted with a hMSC-PA or control, followed by echocardiography analysis. Both ejection fraction (EF) and fractional shortening (FS) were not significantly higher than control group. The data are represented as mean ± SEM. *$p < 0.05$ compared with Sham group; $n = 5$ animals per group. One-way ANOVA was used for statistical analyses. Sham Sham operation, MI CON MI control, PA hMSC-loaded patch

32.21% ± 0.67% in the control group, $n = 7$ animals; $p < 0.05$; one-way ANOVA was used for statistical analyses.) and fractional shortening (hiPSC-CM (CM): 15.02% ± 0.85% vs. hiPSC-CMs + hMSC-PA (CM + PA): 18.61% ± 0.26% vs. 11.56% ± 0.94% in the control group, $n = 7$ animals; $p < 0.05$; one-way ANOVA was used for statistical analyses.) (Fig. 3a). Finally, the therapeutic effects of our complementary approach was directly compared with a dual injection group (Dual-IN), which received an intra-myocardial injection of a cell mixture (hiPSC-CMs, hMSCs, and hdECM). An echocardiography taken at 2, 4, and 8 weeks showed that both EF and FS of the CM + PA group was sig-nificantly higher than the Dual-IN group at both 4 and 8 weeks (Supplementary Fig. 6).

Moreover, histological images demonstrated that the number of capillaries (mm²) in the both border zone and infarct zone of the hearts from the complementary group were substantially higher than the control and the hiPSC-CMs only group (Fig. 3b). The number of functional capillaries (diameter range: 5–10 μm) in the complementary group was significantly higher than other test groups (Supplementary Fig. 7). Importantly, the combined treatment group significantly decreased cardiac fibrosis. The results from Masson's trichrome staining using cardiac tissue harvested at 8 weeks exhibited an area of fibrosis (%) was significantly lower in combined treatment groups compared with other experimental groups (Fig. 3c; Supplementary Fig. 8 and Supplementary Movies 4, 5). Collectively, these results clearly indicate that the combined treatment lead to comprehensive cardiac repair through the improvement of cardiac function, as well as vascular regeneration and reduction of cardiac fibrosis.

**hMSC-PA enhance the engraftment of hiPSC-CMs.** Interest-ingly, we found that implantation of hMSC-PA significantly

improved the retention of intramyocardially injected hiPSC-CMs. The injected hiPSC-CMs were tracked in heart tissues by using two distinct types of hiPSC-CMs: (1) hiPSC-CMs continuously expressing the green fluorescence signal (hiPSC-CMs-GFP) (Supplementary Fig. 9 and Supplementary Movies 6, 2) hiPSC-CMs pre-labeled with a potent red florescence dye, CM-DiI, prior to cell injection (Supplementary Fig. 10). Immunostaining with human-specific antibodies for the MYH7 protein and mito-chondria further verified the identity of hiPSC-CM-GFP as human CMs (Fig. 4a, b). Next, the number of hiPSC-CMs that remained in heart tissues was quantified at different time points by harvesting the samples obtained from two different experi-mental groups (CM vs. CM + PA) at 2, 4, and 8 weeks after administration. Cell counting was conducted manually, and there was significant difference in the number of hiPSC-CMs between the two groups. The CM + PA group displayed a substantially higher count than that of the CM group (Fig. 4c). Notably, the CM + PA group also showed hiPSC-CMs distributed throughout all regions of the left ventricle while the CM group showed hiPSC-CMs only localizing near the injection sites (Supplemen-tary Fig. 10). Both groups were able to retain the majority of hiPSC-CMs within the infarct zone and the cells remained viable until 8 weeks following injection as evidenced by the TUNEL assay (Fig. 4d).

**hMSC-PA improves the maturity of hiPSC-CMs.** Remarkably, immunohistochemistry with TNNT2 and MYH6/7 antibodies revealed that the CM + PA group evidently displayed a more mature form (Fig. 4e; Supplementary Fig. 11). While the mor-phology of hiPSC-CMs in the CM group exhibited a typically immature globular phenotype, the CM + PA group exhibited a much larger rod-shaped structure that resembled adult-like CMs.

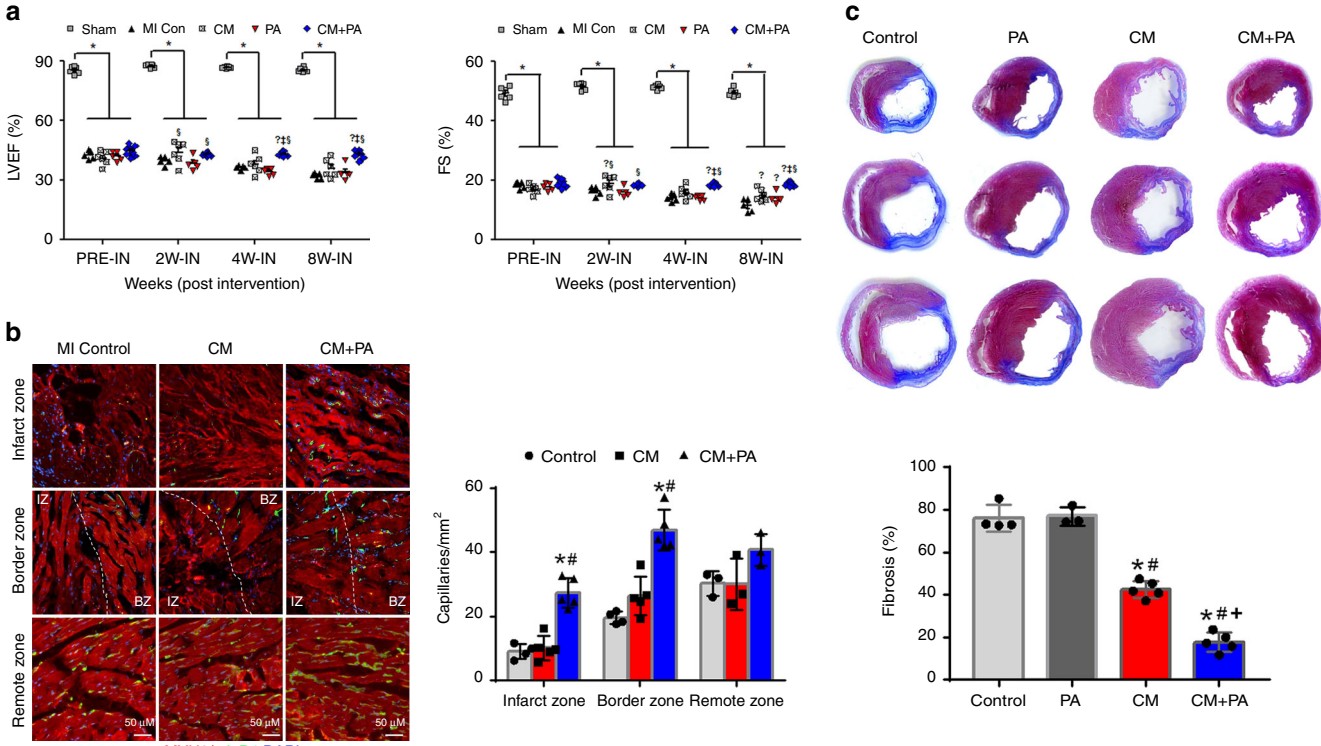

**Fig. 3** Dual approach improved cardiac function, capillary density, and reduced scar formation following MI. **a** Improvement of cardiac function in rats receiving hiPSC-CMs with hMSC-PA. Fractional shortening (FS) and ejection fraction (EF) were significantly higher in the hiPSC-CM with hMSC-PA group compared with other groups measured by echocardiography. The data are represented as mean ± SEM. *$p < 0.05$ compared with Sham group, †$p < 0.05$ compared with MI CON group, ‡$p < 0.05$ compared with CM group, §$p < 0.05$ compared with PA group; $n = 5$ animals per group. One-way ANOVA was used for statistical analyses. **b** Representative images of capillaries stained with FITC-IsB4 (green) on the infarct zone, border zone, and the remote zone at 8 weeks after MI and their quantification summary. For quantification, the number of capillaries on five randomly selected fields in each heart was counted. Scale bars: 100 μm. The data are represented as mean ± SEM. *$p < 0.05$ compared with control, #$p < 0.05$ compared with the CM group; $n = 5$ independent samples per group. One-way ANOVA was used for statistical analyses. **c** Representative images from the four experimental groups showing cardiac fibrosis after staining with Masson's trichrome in the hearts harvested 8 weeks after MI and their quantification results. The data are represented as mean ± SEM. *$p < 0.05$ compared with control, #$p < 0.05$ compared with the PA group, and +$p < 0.05$ compared with the CM group; $n = 5$ biologically independent samples per group. One-way ANOVA was used for statistical analyses. Sham Sham operation, MI CON MI control, CM hiPSC-CMs, PA hMSC-loaded patch, CM + PA hiPSC-CMs + hMSC-loaded patch

These effects were also reflect in vitro when 10 or 30% of hMSC-conditioned media (hMSC-CA) collected from hMSC culture greatly increased the size of cultured neonatal rat ventricular cardiomyocytes (NRVM). The expression level of genes related to CM maturation was not altered in any significant way in exception to cardiac troponin T (TNNT2). Collectively, these results indicate the proteomic milieu secreted by hMSCs are a dominant factor in CM maturation (Supplementary Fig. 12).

More importantly, concomitant staining with Gja1, a major gap junction protein, further showed that a substantial number of engrafted hiPSC-CMs in the CM + PA group formed gap junctions with host CMs (Fig. 4f; Supplementary Fig. 13). Limited expression of Gja1 in the CM group suggests the supportive and essential role of hMSC-PA in the formation of gap junctions between the injected hiPSC-CMs and host CMs (Fig. 4f; Supplementary Fig. 13). In addition, the results from multi-electrode arrays (MEAs) using co-cultured hiPSC-CMs and NRVM in vitro exhibited a well-synchronized action potential, suggesting a well-coupled syncytium between two CMs. No visible sign of arrhythmogenic beating was detected during the recording period (Supplementary Fig. 14 and Supplementary Movie 7). Taken together, these results suggest that hMSC-PA can markedly enhance cell retention, improve functional maturation, and induce integration with the host myocardium in MI hearts.

**Cytokines secreted from hMSCs increases angiogenesis**. To identify detailed therapeutic mechanism of hMSC-PA, we performed various types of in vitro analyses. Among the first, to test whether hMSCs directly promote angiogenesis, we conducted cell migration and tube-formation assays using conditioned media (CM) collected from cultured hMSCs (hMSC-CM). At first, we performed the scratch assay, an in vitro experiment for measuring endothelial cell (ECs) proliferation/migration, which are critical steps of angiogenesis. As shown in Fig. 5a, the addition of 10% of hMSC-CM significantly enhanced the migration of HUVEC and induced faster closure of the cell-free gap compared with the control group, suggesting that cytokines secreted by hMSCs enhanced the mobility of ECs (Fig. 5a). Next, the results from Matrigel tube-formation assay, which was carried out to evaluate vessel-forming capability, showed that tube length and branches assessed 9 h later were significantly higher in the 10% hMSC-CM-treated HUVEC compared with untreated control HUVEC (Fig. 5b).

The effects of hMSCs for EC proliferation were further examined by our co-culture system with hMSCs and HUVEC. As shown in Fig. 5c, the proliferation rate of HUVEC was significantly higher than HUVEC monoculture control group when they were co-cultured with hMSCs (Fig. 5c). On the first day, both the monoculture and co-culture plates expanded at

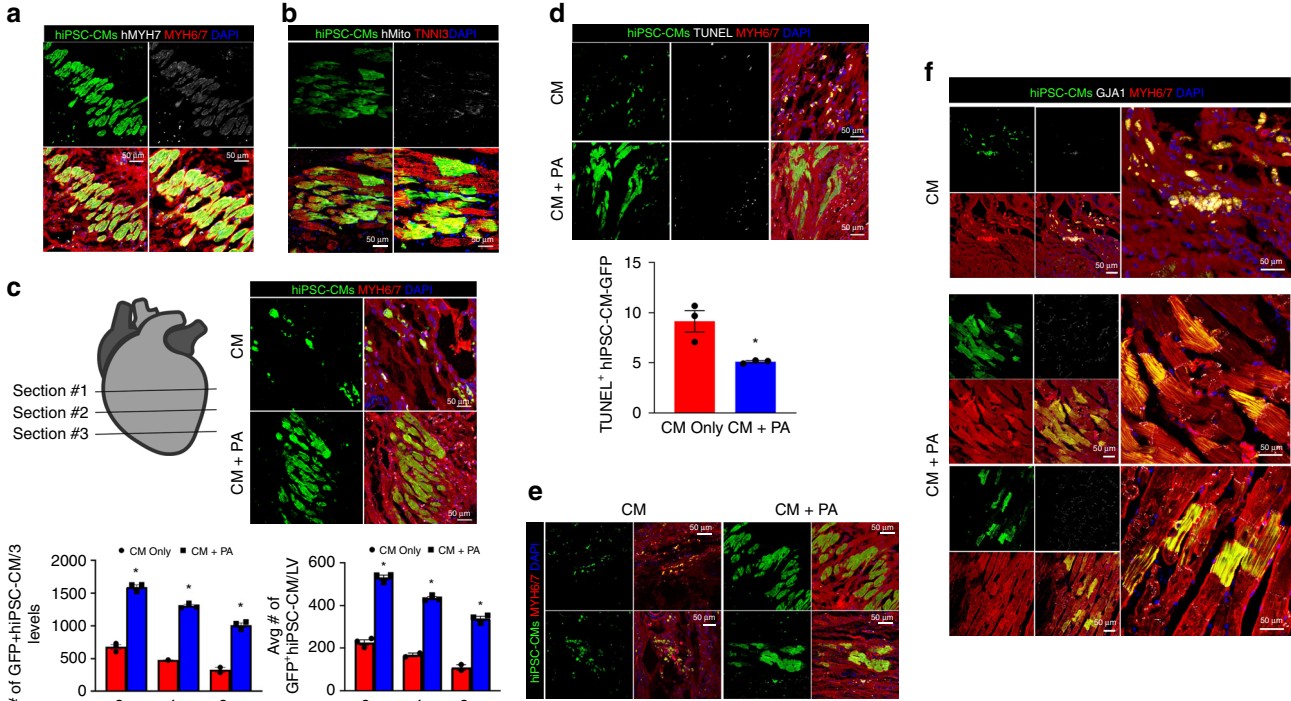

**Fig. 4** Implantation of hMSC-PA improves the retention and maturation of hiPSC-CMs on the infarcted myocardium. **a** Representative immunostaining images with hiPSC-CM-GFP (green), human specific MYH7 (gray), and MYH6/7 (red). Expression of hMYH7 in the hiPSC-CM-GFP verify the identity of hiPSC-CM-GFP as human CMs. Scale bars: 50 μm. **b** Representative immunostaining images with hiPSC-CM-GFP (green), human-specific mitochondria (gray), and MYH6/7 (red). **c** Quantification of hiPSC-CM-GFP remained in the hearts receiving CM or CM + PA. The heart tissue sections at three different locations were prepared and then immunostained with the MYH6/7 antibody. Subsequently, we imaged entire left ventricle (LV) area at three different locations of heart sections and manually counted the hiPSC-CMs positive for both GFP signal (green) and MYH6/7 antibody (red). Representative immunostaining images with hiPSC-CM-GFP (green) and MYH6/7 (red) and their quantification results. The data are represented as mean ± SEM. *$p < 0.05$ compared with CM only group. *T* test was used for statistical analyses. $n = 3$ biologically independent samples per group. **d** Results of TUNEL assay performed to assess the alive hiPSC-CM-GFP in the hearts receiving CM or CM + PA. The data are represented as mean ± SEM. ***$p < 0.05$ compared with CM group; $n = 3$ biologically independent samples per group. *T* test was used for statistical analyses. **e** Representative images of hiPSC-CM-GFP (green) expressing MYH6/7 (red) when they were injected in the absence or presence of hMSC-PA in MI hearts. Scale bars: 50 μm. **f** Representative images of hiPSC-CM-GFP (green) expressing MYH6/7 (red) and GJA1 (gray) when they were injected in the absence or presence of hMSC-PA in MI hearts. Expression of GJA1 indicates integration of implanted hiPSC-CMs with host myocardium. Scale bars: 50 μm. The heart sections were also stained with DAPI (blue) for visualization of nuclei. CM: hiPSC-CMs. CM + PA: hiPSC-CMs + hMSC-loaded patch

roughly the same rate ($\times 10^5$), but a clear difference in growth rate appeared on day 2. The size of HUVEC droplet and cell number in the co-culture plate was markedly higher than those in monoculture of HUVEC, indicating that the factors secreted from hMSCs have significant vasculogenic potential by proliferation and migration of ECs. Collectively, these results indicated that enhanced vessel formation shown in a number of in vitro assays was due to the angiogenic factors secreted from hMSCs, which is consistent with our in vivo results.

**hMSC-secreted factors improve the survival of hiPSC-CMs.** Given the ability of hMSC-PA to improve the survival and retention of injected hiPSC-CMs, we sought to examine whether hMSC-CMs exerted direct cytoprotective effects in hiPSC-CMs in vitro. Ischemic injury was simulated by exposing hiPSC-CMs to $H_2O_2$. Administration of hMSC-CMs significantly improved cell viability as determined by the Annexin V assay and lactate dehydrogenase (LDH) release. As shown in Fig. 6, treatment with 10% of hMSC-CM substantially decreased the number of both PI and Annexin-positive cells (Fig. 6a). Furthermore, the results from LDH assays revealed that LDH released from hMSC-CM-treated hiPSC-CMs was significantly lower than untreated control hiPSC-CMs, suggesting hMSC-CM possess cardioprotective effects against

ischemic insults (Fig. 6b). We also observed that cytokines released by hMSCs were able to induce CM migration in a chemotactic manner as hiPSC-CMs began to migrate towards hMSCs after 12 h and reached the center of the dish at 72 h. Within 48 h, hiPSC-CMs (black arrow) continued to approach the periphery of hMSCs which is indicative of cell–cell affinity (Supplementary Fig. 15).

**Cytokine array in BM-MSC conditioned medium.** Lastly, to identify the panel of factors secreted by hMSCs, we prepared the hMSC-CM from hMSC cultures at two different time points at day 7 and 14 and analyzed them via Proteome Profiler Human Arrays (Fig. 7)[12,13]. The following families of cytokines were observed from hMSC-CM; angiogenesis (angiopietin 1, vasorin, progranulin, IGFBP-2, IGFBP-7, VEGF, DKK-1, DKK-3, IL-6, IL-8, uPA), ECM remodeling (MMP-1, MMP-13, MMP-20, thrombospondin-1, TIMP-1, TIMP-2, TIMP-3, latent TGF-beta, bp1), cell viability (EDA-A2, GDF-15, IL-1 sRII, MCP-1, MIP-2), and inflammation (IL-28A, lymphotactin, activin A, GRO) (Fig. 7).

## Discussion

In this study, we demonstrate a approach in treating MI that exploits the providential advantages of both hiPSC-CMs and

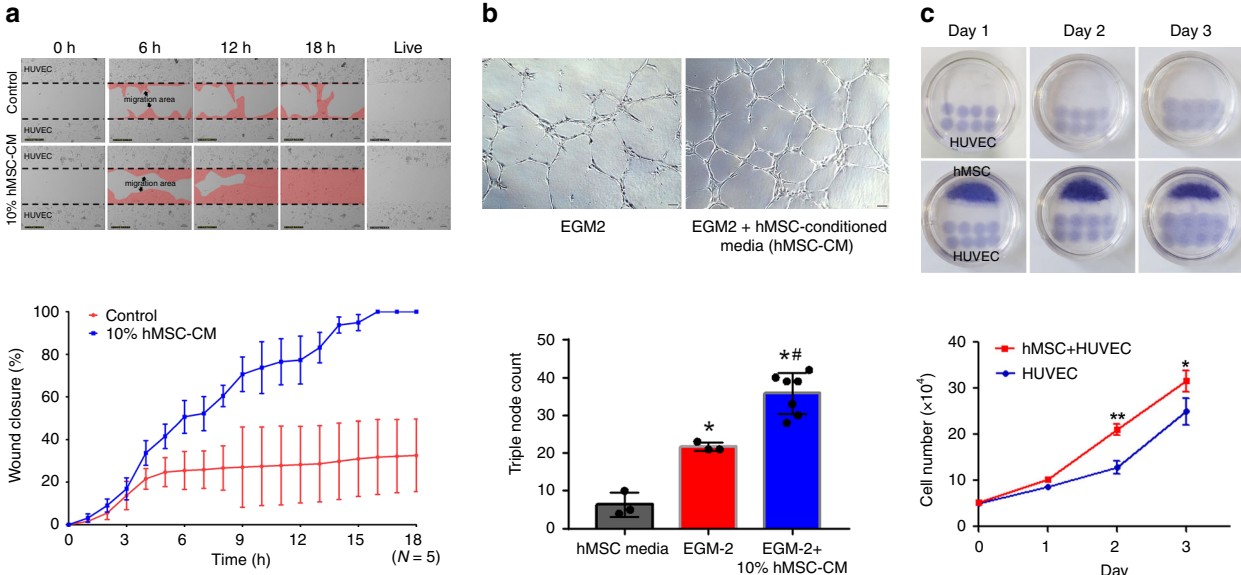

**Fig. 5** Secreted factors from hMSCs promote endothelial cell migration and vasculogenic potential. **a** Endothelial cell-migration assay. The HUVEC were incubated with 10% conditioned media harvested from hMSC cultures (hMSC-CM) or control media to evaluate the migration capability. Representative Images under an inverted microscope and quantification summary. The data are represented as mean ± SEM. *$p < 0.05$ compared with control group; $n = 5$ biologically independent samples per group. T test was used for statistical analyses. **b** Matrigel plug assay. The 10% hMSC-CM was treated to HUVEC on matrigel to examine the vasculogenic potential. Representative images of tubes formed on Matrigel and quantification summary. The data are represented as mean ± SEM. *$p < 0.05$ compared with hMSC basal media, #$p < 0.05$ compared with EGM2 group; $n = 3$ biologically independent samples per group. One-way ANOVA was used for statistical analyses. **c** Increased proliferation of HUVEC when they were co-cultured with hMSCs. Representative images of co-cultures of hMSCs and HUVEC and quantification summary. The data are represented as mean ± SEM. *$p < 0.05$ and **$p < 0.01$ compared with HUVEC only group; $n = 3$ biologically independent samples per group. T test was used for statistical analyses

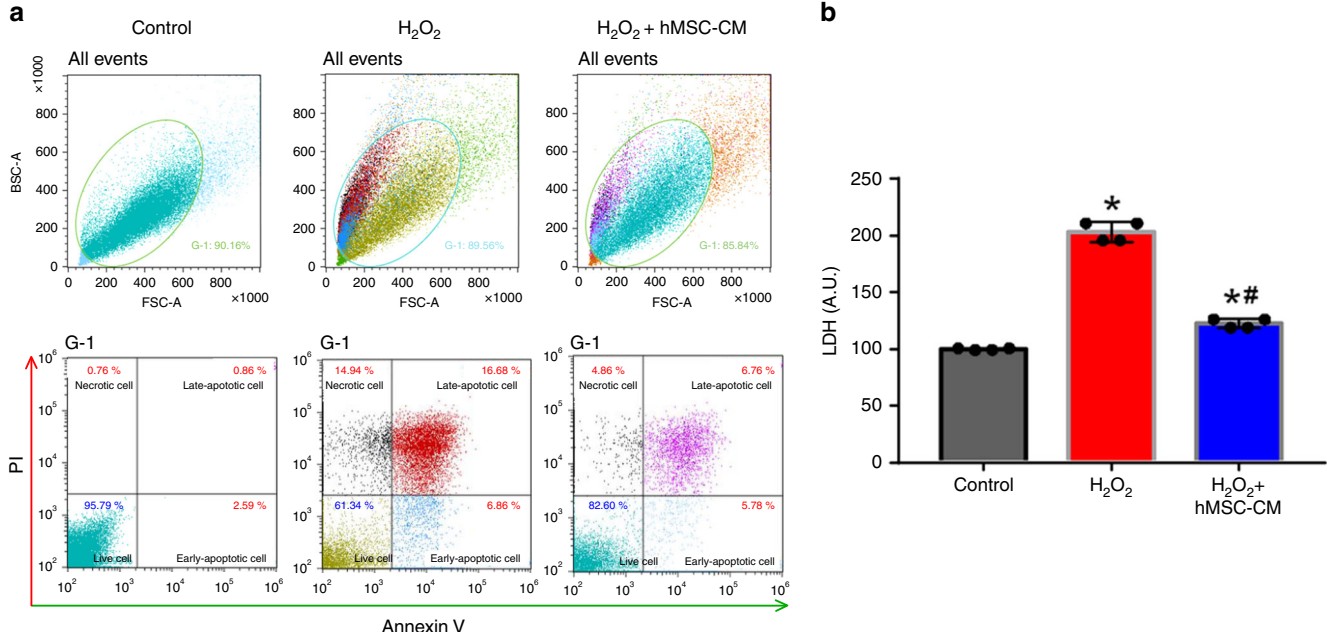

**Fig. 6** Direct cytoprotective effects of the hMSC-conditioned medium on hiPSC-CMs undergoing simulated ischemic injury. **a, b** Treatment with hMSC-conditioned media (hMSC-CM) increased cell survival after $H_2O_2$ (200 μM) treatment as determined by the (**a**) Annexin V and (**b**) lactate dehydrogenase (LDH) assay. The data are represented as mean ± SEM. *$p < 0.05$ compared with untreated control group, #$p < 0.05$ compared with $H_2O_2$-only treated control group; $n = 3$ biologically independent samples per group. One-way ANOVA was used for statistical analyses

hMSC-PA to significantly amplify cardiac repair. Synergistic effects of intramyocardially injected hiPSC-CMs and epicardially implanted hMSC-PA collectively rejuvenated the myocardium and vessels post MI. Epicardially implanted hMSC-PA provided a complimentary microenvironment which enhanced vascular regeneration through prolonged secretion of beneficial paracrine factors as expected, but more importantly it improved the retention, distribution, engraftment, and maturation of hiPSC-CMs which ultimately augmented heart function and restored the injured myocardium (Fig. 8).

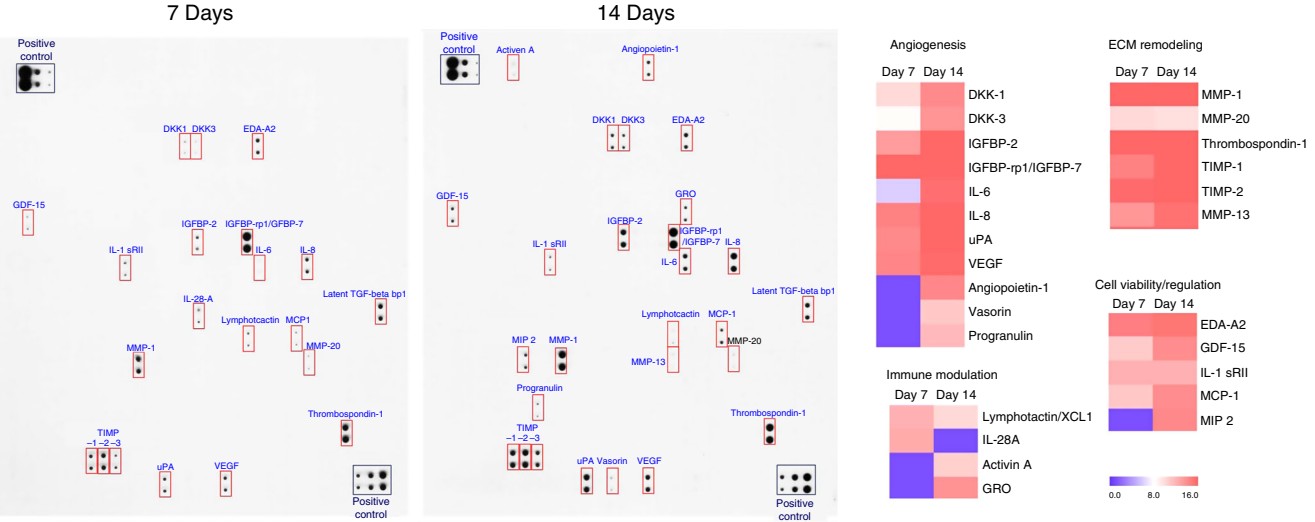

**Fig. 7** Characterization of conditioned media harvested from hMSC cultures by antibody arrays. Multiple array membrane incubated with hMSC-CM collected at day 7 and day 14 reveal the presence of several paracrine factors. hMSC-CM conditioned media harvested from hMSC cultures

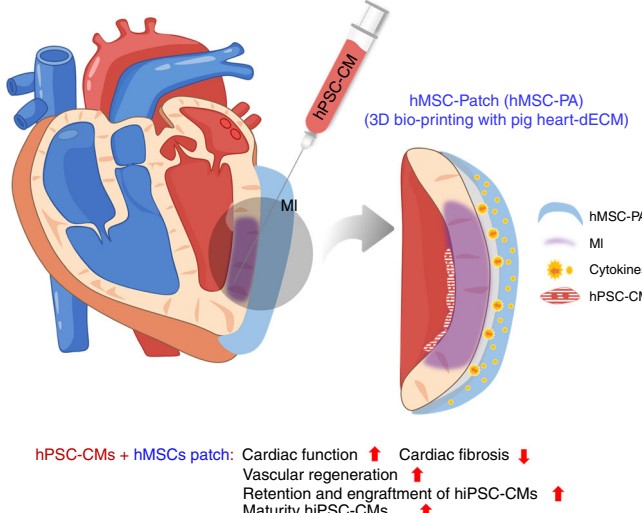

**Fig. 8** Schematic diagram of the underlying mechanism of dual treatment approach of hPSC-CMs and hMSC-patch

Indeed, there have been several previous studies that describe the beneficial effects of hPSC-CMs[5–8] or hMSCs[14–16] or other cell types, such as cardiac progenitor cells[17,18], endothelial cells[19–21], and smooth muscle cells[20,21], on MI either separately or as a combinatory[20–22]. These individual type of cells or cell mixtures were delivered to the hearts through direct intramyocardial injection[5–8] or as a patch form[19,21,23] formulated by using several different types of biomaterials. Compared with those previous reports, this study is, to the best of our knowledge, the first to simultaneously examine the effects of two distinct major stem cell types delivered via two different routes for inducing comprehensive cardiac repair.

A primary obstacle to cell-based cardiac therapy is the extremely low rate of retention and engraftment to the host myocardium which is particularly important because degree of cell-based cardiac repair largely depends on the number of cells that survive and engraft within the heart[24–26]. In this regard, our histological results revealed a substantial increase in the retention and thereby engraftment of intramyocardially injected hiPSC-CMs when paired with epicardial hMSC-PA. The absence of hMSC-PA resulted in a rapid decline of the hiPSC-CM

population over time to dwindling levels by 8 weeks, whereas its presence greatly bolstered the number of survivals for subsequent engraftment (Fig. 4c). From this, we presumed that the paracrine factors secreted by hMSC-PA improved the survival and engraftment of hiPSC-CMs by enabling the cells to more resistant to the hostile microenvironment in ischemic tissues, particularly during the early stage of implantation, which is critical for cell engraftment and survival. We also envisioned that the placement hMSC-PA over the epicardium may have also served as a bio-physical barrier preventing the mechanical expulsion of injected hiPSC-CMs into the epicardial space, further contributing to the increased retention rate of hiPSC-CMs.

Notably, we found that hMSC-PA promoted the maturation of injected hiPSC-CMs in MI hearts. Our histological analyses results demonstrated that hiPSC-CMs together with hMSC-PA led to a more elongated and rectangular cell shape, which are typical morphological characteristics of matured adult CMs. Those hiPSC-CMs stained more strongly for CM-specific MYH6/7 and TNNT2, and exhibited striations with Z-bands running longitudinally along the hiPSC-CMs. They were also more compact and better oriented with adherent intercalated discs. More importantly, they showed significant expression of gap junctions, GJA1, which allows for more efficient electrical and mechanical integration of the host myocardium (Supplementary Fig. 14). In contrast, hiPSC-CMs injected without hMSC-PA showed an immature morphology and expressed minimal levels of GJA1. Thus, it seemed to be that the secreted factors released from hMSC-PA provided the cues necessary for enhancing CM maturation (Supplementary Fig. 12). There is increasing recognition that hPSC-CMs represent immature CMs at an embryonic or fetal stage and are functionally and structurally different from mature CMs[27]. It has been shown that hPSC-CMs at day 30 post differentiation are still small and have disarrayed myofibrils[28]. Hence, these underdeveloped characteristics of hPSC-CMs may significantly limit their usage for further applications. Performing additional studies to identify a secreted factor(s) from hMSCs that can promote the maturation of hPSC-CMs are highly required.

Even though this study evidenced several promising results, there are some limitations to the study that require consideration. Since our complementary approach (CM + PA) was examined in a permanent ischemia model, the outcomes may be different if applied to models with advanced heart failure. In addition, the majority of cardiac imaging results was solely obtained by

echocardiography in this study. Future studies employing more advanced cardiac imaging methods such as Magnetic resonance imaging (MRI) and PET imaging will warrant more accurate and sophisticated cardiac analyses. Lastly, this approach necessitates somewhat complicated technical and surgical procedures to successfully inject hiPSC-CMs and implant hMSC-PA to the heart. Follow-up studies should be carried out to develop a more concise surgical method that can take full advantage of this approach with less invasive procedures.

In summary, we report a strategy for cardiac repair which can concurrently rejuvenate both the myocardium and vasculatures through two major types of stem cells.

Epicardial patch carrying hMSCs improved vascular regeneration and promoted engraftment and viability of the injected hiPSC-CMs leading to subsequent restoration of cardiac function. Our study highlights the unique advantages of different cell types, and their appropriate use can significantly advance cell-based cardiac therapy.

## Methods

**Mesenchymal stem cells derived from human bone marrow**. Human mesenchymal stem cells derived from the bone marrow (hMSCs; Catholic MASTER Cells) were obtained from Catholic Institute of Cell Therapy (CIC, Seoul, Korea). Human bone marrow aspirates were obtained from the iliac crest of healthy donors aged 20 to 55 years after approval by the Institutional Review Board of Seoul St. Mary's Hospital (approval numbers KIRB-00344–009 and KIRB-00362–006). We have complied with all relevant ethical regulations for work with human participants and obtained informed consent. Bone marrow aspirated from each consented donor was collected and sent to the GMP-compliant facility of Catholic Institute of Cell Therapy (Seoul, Korea, http://www.cic.re.kr) for the isolation, expansion, and quality control of hMSCs. The marrow mixture was centrifuged at 4 °C, 793 g for 7 min to obtain a marrow pellet. After removal of the supernatant, red blood cells were removed by adding and suspending in tenfold volume of sterile distilled water. Cell pellet obtained by centrifuging the RBC-deprived sample, was then suspended in the MSC growth medium (Dulbecco's modified Eagle's medium-low glucose (DMEM-LG, PAA), 10% fetal bovine serum (FBS, Gibco). They were added to 100 -mm tissue culture dish (TTC), which was placed in a $CO_2$ incubator to initiate culture. The incubator was maintained at 37 °C with 5% $CO_2$. The MSC growth medium was used for all cell expansion procedures, unless mentioned otherwise. Media were replaced twice per week. Cells were detached when they reached 70–90% confluence and replated at a density of $5–8 \times 10^3$ cells/cm². Cells were expanded 2 to 4 passage in the GMP-compliant facility. During cell expansion, cells were tested for bacterial sterility, mycoplasma sterility, and endotoxin level (<3 EU/mL). In addition, multi-differentiation potential and cellular surface antigens (CD90/CD73, >95% positive; CD34/CD45, >95% negative) were tested for cells after 4th passage.

**Manufacturing hMSC-PA using hdECM**. The heart-derived decellularized extracellular matrix (hdECM) isolated from porcine heart tissues was prepared by following methods that we originally reported[29]. The heart tissue from a 6-month-old Korean domestic pig was purchased from a local livestock product market in Korea[29] and we dissected the left ventricle to cut into small pieces. Then the small pieces of heart tissues were soaked in a 1% sodium dodecyl sulfate (Affimetrix, CA) solution for 48 h followed by treatment with 1% triton X-100 solution in PBS (Biosesang, Korea) for 1 h. Next, the decellularized tissues were dipped in the PBS for 3 days to remove the residual detergent. Subsequently, the decellularized heart tissues were lyophilized, pulverized in liquid nitrogen, and digested in 10 mL of 0.5 M acetic acid solution (Merck Millipore, Billerica, MA) at a final concentration of 3.3 w/v% (330 mg of hdECM powder) supplemented with 33 mg of pepsin powder. The digested hdECM solution was filtered through a 40 -µm pore mesh, aliquoted in 1 ml, and stored at −20 °C for further experiments. Before manufacturing the hMSC-PA, the hdECM solution was adjusted to a neutral pH of 7.4 by adding 10 N NaOH solution, while keeping the conical tube in an ice bucket to avoid gelation of hdECM.

To generate hMSC-PA, we first produced a disk-shaped polycaprolactone (PCL) with 8 -mm diameter and 0.5- mm height as a supporting framework by using a 3D printer according to the generated code (Fig. 1b). Then, we built up the bioink by mixing the hMSCs ($1 \times 10^6$/ml), hdECM (20 mg/ml), and 0.02% (w/v) of vitamin B2 and applied the bioink on the PCL framework generated by 3D printing (Fig. 1c). Subsequently, hMSCs-PA was exposed to UVA light for about 60 s to initiate a vitamin B2-induced post-cross-linking process (light intensity: 30 mW/cm²) and kept them at 37 °C for 24 h for additional thermal cross-linking. After all these procedures, final thickness and diameters of hMSC-PA were averaged 3 mm and 8 mm, respectively.

**Generation of cardiomyocytes from hiPSCs**. hiPSCs (DF19–9–11T)[30,31] cell line was purchased from WiCell® and maintained on Matrigel using the mTeSR1 (STEMCELL Technologies) medium after approval from the Institutional Review Board of Seoul St. Mary's Hospital (approval number KIRB-0050210–007). Subsequently, to initiate the differentiation into cardiac lineage, hiPSCs were then seeded onto a hPSC qualified Matrigel (Corning)-coated cell culture dish (Eppendorf) at 140,000 cells/cm² dish. 5 µM Y-27632 (Tocris) was added for the first 24 h after passage. The medium was changed daily, and hiPSCs were allowed to grow in mTeSR1 for 3–4 days until the cells were 90% confluent. At day 0, cells were treated with CHIR99021 (Tocris) 8 µM/ml in CDM (cardiomyocyte differentiation medium: RPMI1640 (ThermoFisher Scientific)/BSA (Sigma-Aldrich)/ascrobic acid (Sigma-Aldrich). After 48 h, the medium was changed to CDM supplemented with 3 µM/ mC C59 (Wnt inhibitor/Stemgent Inc.) for another 48 h[9–11]. At day 5, the medium was replaced with CDM and freshly changed every 2 days. Spontaneously, contracting cells began to appear at ~day 6 to day 10. From day 10 to day 15, the medium was replaced with CDM containing L-lactic acid to metabolically select and purify cardiomyocytes (hiPSC-CMs)[32]. All live images were taken with a Lumasope 720 microscope (Etaluma).

**Generation of hiPSC-CM-GFP**. hiPSCs expressing GFP signal (hiPSC-GFP) was generated by using pMXs-EGFP retroviral vector. Briefly, the pMXs-EGFP retroviral vector was purchased from Addgene and transfected into the 293FT cells using the chemical method, Lipofectamine 2000 (Invitrogen). After 48 h, the virus-containing supernatants were collected, filtered through a 0.45-µm Milex-HV filter (Millipore), and then concentrated using Retroconcentin (SBI; Mountain View, CA). Subsequently, these were transfected to $5.0 \times 10^5$ of hiPSC (DF19–9–11T) in opti-MEM media (Life Technologies) supplemented with 4 mg/mL polybrene (Millipore). Finally, hiPSC colonies expressing GFP signal were FACS sorted based on the expression of EGFP at 3–4 days post retroviral transduction (Supplementary Fig. 9). The purified fraction of hIPSC-GFP colonies were expanded for further differentiation into the CMs. Finally, hiPSC-GFP were differentiated into the hiPSC-CMs-GFP through the previously used CM differentiation protocol.

**Myocardial infarction model and cell/patch delivery**. All animal studies were approved by Animal Care and Use Committee in the Catholic University of Korea. We have complied with all relevant ethical regulations for animal testing and research. Fischer 344 rats (180–200 g, male, Orientbio, Korea) were anesthetized with 2% inhaled isoflurane and intubated via the trachea with an 18-gauge intra-venous catheter. The rats were then mechanically ventilated with medical grade oxygen. Animals were placed on a 37 °C heating pad to prevent cooling during procedure. After shaving the chest, a left thoracotomy was performed. MI was achieved by tying a suture with sterile polyethylene glycol tubing (22 G) placed into the left anterior descending (LAD) artery for 1 min, and then the knot was permanently ligated using a 7–0 prolene suture. To establish baseline left ventricular function, the ejection fraction (EF) and regional wall motion abnormalities (RWMA) were examined post operation day (POD) 7 (inclusion criterion: EF <45% by echocardiographic evaluation). On the same day, rats were anesthetized again using isoflurane inhalation, intubated and mechanically ventilated. The animal chest was re-opened, and the pericardium was partially removed from the infarcted heart. Then, hiPSC-CMs ($1.0 \times 10^6$ per rat) were injected at two different sites in the border zone of infarcted myocardium and the hMSC-PA was implanted directly on the epicardium using two sutures (Supplementary Movie 3). To trace the injected hiPSC-CMs within the heart tissues, we used CM-DiI (Chloromethlybenzamido, CellTracker ™). The stock solution was prepared using a modification of the manufacturer's instructions. From a 1 mg/ml CM-DiI stock solution in DMSO, 5 µM solutions were made in 500 µl of the DMEM and this working solution was used to label the hiPSC-CMs. The chest was closed aseptically, and antibiotics and 0.9% normal saline solution was given. All rats received following immunosuppressants (azathioprine, 2 mg/kg; cyclosporine A, 5 mg/kg; methylprednisolone, 5 mg/kg) daily.

**Echocardiography**. The assessment of functional improvement for injured cardiac tissues was performed with echocardiography[26,33,34]. The rats were lightly anesthetized with inhaled isoflurane, and physiological data were recorded by using Transthoracic echocardiography system equipped with a 15 MHz L15–7io linear transducer (Affniti 50 G, Philips). Serial echocardiograms were performed at 2, 4, and 8 weeks after treatment. The echocardiography operator was blinded to the group allocation during the experiment. Ejection fraction (EF) and fractional shortening (FS), which are indexes of LV systolic function, were calculated with the following equations, respectively:

$$EF(\%) = \left[ \left( LVEFDD^3 - LVESD^3 \right) / LVEDD^3 \right] * 100 \qquad (1)$$

$$FS(\%) = \left[ \left( LVEDD - LVESD \right) / LVEDD \right] * 100 \qquad (2)$$

**Flow-cytometry analysis**. hMSCs were re-suspended in 100 µl of FACS solution (1% FBS in PBS) then incubated with PE-conjugated mouse anti-human 1:250 CD90 (BD Biosciences # 555596), 1:250 CD73 (BD Biosciences # 550257), 1:250 CD105 (BD Biosciences # 560839), and APC-conjugated mouse anti-human 1:250

CD44 (R&D System, FAB4948A). In case of hiPSC-CMs, hiPSC-CMs were first washed twice with PBS for 1 min at RT followed by 0.05% Trypsin-EDTA (thermos) for 3–4 min at 37 °C. After re-suspension and fixation with PFA/permeabilized solution (BD Bioscience) for 15 min and stained using 1:700 TNNT2 (Cardiac troponin T; Thermo Fisher #MA5–12960) and ACTN2 (α-sarcomeric actinin; Sigma-aldrich #A7811) IgG antibodies for 12 h at 4 °C. Secondary staining was performed with antibody 1:1000 Alexa Fluor 488 goat anti-rabbit IgG (Thermo Fisher #A11008) and 1:1000 Alexa Fluor 555 goat anti-mouse IgG (Thermo Fisher #A21422) antibodies for 1 h at RT. All cells were analyzed using a FACSCalibur, Cell Quest software (BD Biosciences) with the exception of hiPSC-GFP which was sorted through a SH800S Cell Sorter flow cytometer with Cell sorter software Ver 2.1.2 (Sony Biotechnology).

**Quantitative real-time RT-PCR**. The total RNAs were extracted by the addition of 0.5 mL of TRIzol reagent (Life Technologies) to cells on a plate as described in the manufacturer's instructions. One microgram of RNA was subjected to cDNA synthesis with SuperScriptTM Reverse Transcriptase IV and random primers (Invitrogen). SYBR® Green PCR Master Mix (Applied Biosystems) was used to detect the accumulation of PCR product during cycling with the ABI Real-time PCR StepOne Plus (Applied Biosystems). Real-time reverse transcription-polymerase chain reaction (RT-PCR) was carried out in triplicate in at least three independent experiments. Oligonucleotide primers were designed using real-time RT-PCR system sequence detection software v2.3 (Applied Biosystems), and their sequences are provided in Supplementary Table 1. Fold differences in the expression level of each gene were calculated for each treatment group using CT values normalized to transcript levels of the housekeeping gene, 18S rRNA or GAPDH, according to the manufacturer's instructions.

**Capillary density measurement**. At the time of sacrifice, hearts were perfused with GFP-conjugated Isolectin B4 from Griffonia simplicifolia (Vector Lab) for 15 min at room temperature. Hearts were then fixed in 4% paraformaldehyde overnight before embedding in OCT compound (Thermo Scientific) with dry ice. In all, 10- μm cross-sections of the heart were made by using HM525 NX Cryostat (Thermo Scientific) starting from the apex to top. The sections were stored in −80 ºC before use. The number of capillaries were counted in five random microscopic fields using a fluorescence microscope (Nikon) and expressed as the number of capillaries per square millimeter tissue.

**Determination of fibrosis**. Masson's Trichrome (MT) staining (Sigma) was performed to determine the fibrosis area of MI hearts. Briefly, three frozen sections were fixed in Bouin's solution at 56 °C for 15 min in each group. These sections were stained using Weigert's iron hematoxylin solution for 5 min at room temperature and also stained using Biebrich Scarlet-acid Fuchsin solution for 2 min at room temperature. Finally, the sections were counterstained with Aniline Blue for 5 min, followed by incubation in 1% acetic acid for 2 min at room temperature. Extensive washes were performed between each step. The collagen fibers appeared blue and viable myocardium appeared red. The percent of the area of fibrosis to entire left ventricular wall area was quantified using ImageJ software with basic add-ons.

**Immunocytochemistry**. Cells were plated onto gelatin-coated glass dish and cultured for 5 days. Then cells were fixed with 4% PFA for 20 min at 4 °C, permeabilized with 0.1% BSA in 0.03% Triton X-100 for 10 min at room temperature (RT), and blocked in 10% normal goat serum (NGS, ThermoFisher Scientific) in 0.03% Triton X-100 for 30 min at RT. Subsequently, the cells were stained with 1:100 CD90 (Abcam #Ab225), 1:500 TNNT2 (Thermo Fisher #MA5–12960), and 1:200 ACTN4 (Sigma-aldrich #A7811) overnight at 4 °C in 0.03% Triton X-100. Cell were washed three times for 10 min, with 0.03% Triton X-100 and incubated for 1 h at RT in the dark with secondary antibody 1:1000 Alexa Fluor 488 goat anti-rabbit IgG (ThermoFisher Scientific) and Alexa Fluor 555 goat anti-mouse IgG (ThermoFisher Scientific) on a shaker. Cells were washed four times before nuclei were stained with DAPI (Thermo Fisher). All images were analyzed using a fluorescence microscope, Nikon TE2000-U (Nikon, Japan).

**Immunohistochemistry**. Immunofluorescence was performed on 10-μm-thick sections. Following permeabilization with PBS containing 0.5% Triton X-100 for 15 min and blocking with 1% BSA in PBS for 60 min at room temperature, the sections were incubated with primary antibodies diluted in PBS containing 1% BSA and 1% Tween 20 at 4 °C overnight. Primary antibodies used in this study include 1:100 CD90 (Abcam #Ab225), 1:100 MYH6/7 (Abcam #Ab50967), 1:100 TNNT2 (Thermo Fisher #MA5–12960), 1:200 ACTN4 (Sigma-Aldrich #A7811), 1:100 GJA1 (Abcam #Ab11370), 1:100 human MYH7 (Abcam #Ab172967), and 1:100 human mitochondria (Abcam #Ab92824). After washing three times with 1% Tween 20 in PBS, the samples were incubated with secondary antibodies for 60 min at room temperature in the dark. Secondary antibodies used in this study include either 1:400 anti-mouse IgG Alexa Fluor 488 (Invitrogen #A10680) or 1:400 anti-rabbit IgG Alexa Fluor 647 (Invitrogen #A21245). After washing again with 1% Tween 20 in PBS, the sections were stained with DAPI solution (VectaShield) for nuclear staining and then mounted on slides. Imaging of heart sections was performed with a Laser Scanning Microscope LSM 880 NLO with Airyscan processing (Zeiss).

**Multi-electrode array measurements**. In vitro co-cultures of NRVM and hiPSC-CM were prepared. NRVMs and hiPSC-CMs were mixed at a ratio of 8 to 2 and seeded to a multi-electrode array (MEA) chamber (60MEA200, Multi Channel Systems, Reutlingen, Germany). Electrophysiological recording was performed 7 days after seeding. During the measurement, CM was placed in the MEA recording apparatus (MEA2100-System, Multi Channel Systems) maintained at 37 °C and 5% CO$_2$. Spontaneous electrical activity of CM was recorded for 20 min. The recording and analysis of conduction velocity was performed and calculated with Cardio2D program (Multi Channel Systems).

**TUNEL assay**. TUNEL staining was performed via an In situ Apoptosis Detection kit (Invitrogen), as the manufacturer instructed. After TUNEL staining, heart sections were stained with antibodies specific for CMs or hMSCs. Samples were imaged under a confocal microscope, and five views were randomly selected from each section to quantify the number of TUNEL-positive cells.

**Production of hMSC-conditioned medium**. hMSCs ($2 \times 10^6$) were seeded onto 100 -mm dishes, and were cultured until 80–90 % confluency. Cells were then washed with PBS, and the medium was changed to low glucose DMEM (Lonza) without FBS. After 7 of 14 days of culture, the supernatants were collected and kept at 4 °C for further experiments.

**Tube-formation assay**. Basement membrane matrix (Matrigel®, BD Biosciences) was added to two-well chamber slides and solidified by incubation at 37 °C for 30 min. Overall, $1 \times 10^5$ human umbilical vein endothelial cells (HUVECs) were plated onto each Matrigel-containing well with DMEM/F12 medium containing 20% KnockOut serum replacement, 1% nonessential amino acids, 0.1 mM β-mercaptoethanol, 4 ng/mL of FGF2, 10 ng/ml VEGFA, 10 ng/ml EGF, and 25 ng/ml DLL4 and incubated at 37 °C for 12 h. After removing the media, 4% PFA was added for fixation. The tube structures were evaluated by microscopy.

**Endothelial cell-migration assay**. HUVEC ($3.5 \times 10^4$ cells/well) were cultured in a well separated by a Culture-Insert 24 (ibidi, Martinsried, Germany). Next, to record the migration of cells to the cell-free gap, the culture inserts were removed after 24 h and cultured in EBM-2 media (Lonza) supplemented with 10% hMSC-conditioned media for another 18 h[35]. Subsequently, the migration area was digitally photographed during the indicated hours (0, 6, 12, and 18 h) and area of the cell-free gap was calculated by using ImageJ software (version 6.0, NIH). Migration rate was determined and expressed as a percentage of closure relative to the initial size at 0 h[36]. All images were taken with a Lumasope 720 microscope (Etaluma) in time-lapse at 5 min intervals.

**Endothelial cell proliferation assay**. A co-culture system involving hMSCs and HUVECs (Lonza) was used to evaluate the proliferation of HUVECs. The proliferation rate of HUVEC droplets ($1 \times 10^4$ cells; 10 μl) co-cultured with BM-MSCs were measured[12]. Briefly, a total of $2 \times 10^5$ hMSCs were seeded in the upper layer of 35 -mm plates and cultured with 2% FBS in the DMEM medium. After 4 h, HUVEC droplets ($1 \times 10^4$ cells; 10 μl) were carefully placed into the bottom layer of each well. The HUVEC droplets were allowed to adhere at 37 °C for 3 h followed by the addition of medium. Subsequently, H&E staining was carried out to measure the size of HUVEC droplet. Lastly, the total number of HUVECs per plate was quantified. Both HUVEC droplet size and cell number were quantified for 3 consecutive days.

**Cytoprotective effects of hMSCs conditioned medium**. To investigate whether hMSCs conditioned medium could provide cytoprotective effects against ischemic insult, 10% hMSC-conditioned medium was added into hiPSC-CM culture containing hydrogen peroxide (H$_2$O$_2$) (200 μM), simulating conditions of myocardial ischemia in vitro. Following 2 h exposure to H$_2$O$_2$, both hiPSC-CMs and culture medium were harvested. First, hiPSC-CMs were examined by using Annexin V–FITC kit (Biolegend) to measure the apoptosis, and the results were analyzed by SONY® flow cytometer (SH800, Sony Biotechnology, Inc., Tokyo Japan) with sony software 2.1.2version. In addition, lactase dehydrogenase (LDH) assay (LDH cytotoxicity assay kit, Sigma) was performed by using culture medium to measure cellular damage. Briefly, 50 μL of culture medium was collected from various groups of hiPSC-CM cultures (in triplicate) and transferred into a 96-well plate. After adding an equivalent volume of LDH reagent into each well, the absorbance was measured using a spectrophotometer at a wavelength of 492 nm with a reference wavelength of 620 nm. For more accurate measurements, the absorbance of the no-cell controls was subtracted from the readings of hiPSC-CM samples.

**Cardiomyocyte migration assay**. hiPSC-CMs and hMSCs were cultured on Culture-Insert 2 well (ibidi) at $2 \times 10^4$ cells/well. The culture inserts were removed

after 36 h. Both cell types were cultured in the RPMI medium supplemented with 2% FBS for 48 h at 37 °C. The migration area was digitally photographed at the indicated hours (0, 12, 24, and 48 h). All images were taken with a Lumasope 720 microscope (Etaluma) in time-lapse videos at 3 min intervals.

**Cytokine array**. hMSC-conditioned medium was analyzed by using biotin label-based human antibody array I membrane for 507 human proteins (RayBiotech, Norcross, GA)[13,37].

**Statistical analysis**. All quantitative data are shown as means ± SE unless otherwise indicated. The statistical differences between two groups were analyzed by two-tailed Student's *t* tests. The Statistical differences among three or more groups were analyzed by one-way ANOVA with Bonferroni's post hoc analysis. The results were considered statistically significant when the *p*-value was less than 0.05.

## Data availability
The data that support the findings of this study are available from the corresponding authors upon reasonable request.

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

## Acknowledgements
We would like to thank Mr. Joseph Seo at Konkuk University School of Medicine for instrumental discussions, directions and edits for this paper. This study was supported by CityU Research Project (9610355 to K.B.), the Hong Kong Research Grants Council (21100818 to K.B.), National Research Foundation of Korea grants (2015-M3A9C7030091, 2016R1C1B2015529 to S.H.M. and H.J.P., respectively), and the Bio & Medical Technology Development Program grant (NRF-2017M3A9B3061954 to H.J.P.) funded by the Ministry of Science & I.C.T. This study was also supported by the Technology Innovation Program funded (Grant No. 20000325 to S.H.M.) from the Ministry of Trade, Industry & Energy (MOTIE), Republic of Korea.

## Author contributions
S.J.P.: experimental conception and design, acquisition of data, analysis and interpretation of data, paper drafting and revising; R.K.: experimental conception and design, acquisition of data, analysis and interpretation of data, paper drafting and revising; B.W.P.: experimental conception and design, acquisition of data, analysis, and interpretation of data; S.L.: experimental conception and design, acquisition of data, analysis and interpretation of data; S.W.C.: experimental conception and design, acquisition of data, analysis, and interpretation of data; J.H.P.: acquisition of data, analysis of data; J.J.C.: acquisition of data, analysis of data; S.W.K.: acquisition of data, analysis of data; J.J.: experimental conception and design; D.W.C.: experimental conception and design, financial support, administrative support; H.M.C.: experimental conception and design, financial support, administrative support; S.H.M.: experimental conception and design, financial support, administrative support, paper drafting and revising, final approval of

paper; K.B.: experimental conception and design, financial support, administrative support, paper drafting and revising, final approval of paper; and H.J.P.: experimental conception and design, financial support, administrative support, paper drafting and revising, final approval of paper.

## Additional information

**Competing interests:** The authors declare no competing interests.

