## [peer review file · Nature Communications]

Reviewer #1 (Remarks to the Author):

In the manuscript "Dual stem cell therapy synergistically improves cardiac function and vascular regeneration following myocardial infarction", authors aimed to examine the combinatory therapeutic effects of intramyocardially injected human iPSC derived cardiomyocytes and an epicardially attached biomaterial patch. Although adding a cell/tissue patch to rescue myocardial infarction has been reported before, this manuscript provided a very creative innovation of engineering a biomaterial filled patch, using lyophilized porcine heart ECM and hBMC derived MSC as bioink on a 3D printed scaffold. Combinatory cell therapy is a trending topic with numerous studies with the hope to increase cell retention, survival, and engraftment following delivery. Most combinatory cell therapy simply mix multiple stem cell types together prior to delivery in order to increase direct intercellular interaction. Interestingly, This manuscript provides a different way to combine iPSC-CM and MSC, with injecting iPSC-CM (the functional cell) directly into border zone, and patching MSC on top of the infarct area, providing structural and potential vasculature support to the infarct zone. The addition of ECM to the tissue patch can further provide tissue scaffolding to support engraftment. Also, lyophilization provide convenient storage and transportation of cell-free ECM. Therefore, this novel technique may potentially provide translational and clinical benefit to the field of cardiac repair.

The hypothesis and the dual cell type patch-injection method are very interesting and theoretically beneficial. The manuscript attempted to show long term retention of iPSC-CM, and to the engraftment and functional improvements were mostly benefited from MSC's angiogenic potential. Unfortunately, their data fail to support their claims due to flawed experimental design, lack of major controls, and over interpretation of evidence.

Major problems:

1. BMC-MSCs are a widely used and well characterized stem cell type in cardiac injury, with known improvement of cardiac function via fusion with resident cardiomyocytes, transdifferentiation into myocytes or vasculatures. In this manuscript, the authors mainly focused on angiogenesis from MSC. Have the authors examined if any patched MSC migrated into the myocardium? If so, what is the quantity of MSC detected and what lineage they adopted? Have the authors checked how long can MSC survive inside the patch after attachment in vivo (eg. Examine the patch with cell viability markers, proliferation markers, or cell death markers)? Are there vasculature formed inside of the patch?
2. For a direct combinatory approach, have the authors tried directly combining iPSC-CM and MSC in porcine ECM, and inject this dual cell + ECM mixture for MI treatment? And what is the comparison between directly combination vs. using the patch?
3. One of the major claims of this manuscript is "dramatic increase in retention". However, no quantification was provided in in vivo iPSC-CM delivery in order to support this claim. Of 1 million hiPCS-CM delivered, how many CM (with striation morphology) were detected at 2, 4, 8 weeks? Are these cells exclusively from iPSC-CM or could it be migrated MSC from the patch?
4. In order to support increased viability, please provide cell death assessment in in vivo assays including co-immunostaining and quantification with apoptotic/TUNEL markers of iPSC-CM after injection.
5. DiI was used as a cell tracker in intramyocardial injection. Please provide the DiI dilution used methods. DiI is a cell membrane free entry sphere, which does not reflect the host cell morphology. Its strong intensity can shine through neighboring tissue, therefore can make cell source interpretation very confusing with the resident tissue which are not derived from the injected iPSC-CM. DiI cannot be used as the exclusive marker for viable cells. Although authors provided hTNNI3 staining in Fig. 4C, the image quality is too poor to be analyzed. The hTNNI3 pattern does not resemble cardiomyocyte striation as seen in Fig. 4B (CM+PA panels). Please provide better quality images of co-immunostaining of human cardiomyocyte marker or human species marker (HNA, hMito) in addition to DiI to exclusively confirm cell source (Fig 4).
6. It is interesting to see tight junction of iPSC-CM to resident CM. However, the area chosen in Fig 4B are not representative, both CM groups are resident myocyte sparse, whereas both CM+PA groups are with nicely aligned cardiomyocyte. Please provide images of comparable and

representative regions across CM and CM+PA groups. In addition, in order to exclusively confirm human and rat myocyte share tight junction, please provide higher magnification images of GJA1 co-staining with human cardiomyocyte marker and rat myocyte marker on each side of GJA1.

7. In all microscopy images, please provide co-staining with nuclear counterstaining, background tissue staining (eg. myocardium, cytoskeleton, cell boundaries) in order to provide reference landmark for interpretation.
8. The hiPSC-CM contraction seems unsynced. Please provide physiological data to show if hiPSC-CM xenograft caused arrhythmia to host rat heart.
9. In all cardiac function measurements, the basal level (pre) of EF (~40%) or FS (<20%) seemed very low as a healthy normal heart. At post-MI time points, the drop of EF or FS is within ~10% over 8 weeks even in control (no intervention) group. A 10% difference in EF is very minimal as an infarct damage. In fact, the CM+PA kept the EF and FS constant throughout the post-MI time course with no functional compromise at all. This function data is inconsistent with fibrosis staining where 80% of tissue are fibrotic in control animal, but functionally only lost 10% EF (Fig 2B, 3A, 3C). Can the authors comment on this inconsistency? In order to rule out technical inconsistency, please provide representative echo image/video, long/short axis, reference heart rate (RPM), and body temperature data. In all in vivo assays, please provide sham throughout all time points to set a comparison reference.
10. PA only group is missing in Fig 3. Although it was shown in Fig 2, it'll be constructive to see all four group quantified and graphed together for direct comparison.
11. According to figure legend, Fig 2A shows representative area of capillaries in border zone, however the quantification is from infarct zone. In order to claim the increased density of capillaries are exclusively benefit from PA, please provide images and quantification for both border zone and infarct zone in comparison to remote zone (area that has no direct contact with PA). It'll also be helpful if authors can label (with dash lines) bz/iz on images as they did in Fig 4A.
12. In Discussion, authors claimed that the patches can prevent expulsion of injected hiPSC-CM into epicardial space. The suturing of patch seems an invasive procedure to the epicardium. As from the video provided, it seems patch stayed outside of epicardium, which is perhaps ruptured or punctured by suturing. Please specify if the patch is engineered inside or outside of the epicardium?
13. The authors claim that the hiPSC-CM are 98% purity (TNNI3). However, it is surprising to see beating cardiomyocytes migrate towards MSC (sup Fig 5). Why would beating myocytes migrate? Are there any MSC migrate towards myocytes? What is the benefit of a migrating cardiomyocyte? In order to support the claim, please provide migrating assay with fluorescently labelled hiPSC-CM and MSC to separate these two cell types with two colors, or provide live imaging video.

Minor problems:

1. Cardiac patch has been reported before with various cell types (endothelial cells, smooth muscle cells, iPSCs, cardiomyocytes, etc) and binding material (collagen, fibrin, hydrogel, etc). Please cite these publications in your introduction.
2. This study is not "the first to simultaneously examine the effects of two distinct major stem cell types" in cardiac repair. Multiple previous studies reported directly injection of combinatory cell therapy, eg. MSC+CPC (Williams et al, Circulation, 2013), (Quijada et al, Circulation Research, 2015). Please cite these references.
3. It is interesting to see an increased number of capillaries in PA engineered heart. In order to make sure these capillaries are properly functional, please also provide quantification of diameter ranges (Fig 2A, Fig 3B).
4. Please keep experimental procedure to method section. Some content in result section should belong to methods.
5. Please provide size of scale bar.
6. What is the purpose of placing a PE tubing into LAD? And was the tubing ever removed?
7. Please specify the injection volume for 1 million cells.
8. Discussion session is redundant data presentation and too much speculation.
9. Reference 19 and 40 are redundant.

Reviewer #2 (Remarks to the Author):

Kim, et al. present a manuscript showing combination of PSC-CMs and MSC synergistically enhance graft survival in rat myocardial infarction model. They showed hMSC-PAs helped PSC-CMs transplanted directly into infarcted myocardium to retain, survive, and mature through paracrine mechanisms. They also showed angiogenic effects of PSC-CMs both in vitro and in vivo. The concept of the study is interesting and would attract attention; however, there are several significant limitations as follows;

1) The most significant progress in this manuscript, in my view, would be cardiac graft maturation by co-transplantation of hMSC-PAs. Grafted CM shown in Fig. 4 and Supplementary Fig. 4 look very matured CMs which is surprising to me. Are these really graft derived human cells? In fact, these cells strongly expressed MYH6, commonly expressed rodent CMs but not in human. I would like to know if these CM are unequivocally graft-derived cells by following experiments.

a) Please check the expression of MYH7 in graft CMs. In our hands, host (rat) CMs almost exclusively express MYH6 and graft (human) CMs express only MYH7.

b) The authors should try human specific antibody with chromogenic staining to avoid autofluorescence.

c) No attempts have been made to elucidate the mechanisms of cardiac maturation.

d) Likewise experiments in Fig .5, in vitro maturation study would be required.

2) The authors claimed that hMSC-PAs secreted paracrine factors by RT-PCR of hMSC-PA transplanted and non-treated hearts. I feel cell-free patch transplanted control is required.

3) Important information is missing regarding timepoints of histology in Fig. 4.

4) Control study is missing regarding migration study (Supplementary Fig. 5).

5) Some figure numbers in the text do not correspond to the actual figures (e.g. P. 6, line 5).

Reviewer #3 (Remarks to the Author):

The authors aimed to study the combined reparative potential of cardiomyocytes derived from human induced pluripotent stem cells (hiPSC-CMs) and human mesenchymal stem cell-loaded patch, generated from a porcine heart (hMSC-PAs).

Their rationale for this study is that the combination of both hiPSC-CMs and hMSCs-PAs may simultaneously rejuvenate the myocardium and vasculatures.

The authors hypothesized that while intramyocardially injected hiPSC-CMs would restore heart function by engraftment with the host myocardium, epicardially implanted hMSC-PAs would simultaneously enhance vascular regeneration through consistent secretion of angiogenic paracrine factors in MI-induced hearts.

Major findings:

1. Implantation of hMSC-PAs increase the expressions of multiple factors involved in angiogenesis, anti-fibrosis, and anti-inflammation in MI hearts (not a new finding).

2. The dual approach of intramyocardially injected hPSC-CMs and hMSC-PAs improved cardiac function, capillary density and reduced scar formation following MI, compared with hiPSC-CM or hMSC-PAs alone groups.

3. hMSC-PAs enhance the engraftment of intramyocardially injected hiPSC-CMs and improve their maturity compared with the injection of hiPSC-CM alone.

4. Cytokines secreted from hMSCs increases angiogenesis- (not a new finding)

The present work has both strengths and limitations:

Strengths:

The concept is of great interest to the field of cell therapy. However, the idea is not entirely new.(Ichim, Solano, et al. 2010)

Limitations:

1. Some of the major findings in this study are not new as indicated.

2. The authors used human iPSC-CM and human MSCs loaded on a patch generated from a porcine heart in a rat model of MI. The risk of xenogeneic cells rejection is significant.
3. The authors did not evaluate the effect of treatment on cardiac remodeling and the morphometric assessment in figure 3C is minimal.
4. The authors should add characterization of the hMSC-PA in vivo (during MI)- for example, cell survival in the patch, histological staining of the patch.
5. Statistic: changes along time should be tested with repeated measure ANOVA.
6. In figure 2A and 3B, the authors show a higher number of capillaries in the infarct zone of MSC-PAs and MSC-PAs+ iPSC-CM implanted hearts compared with control. This could be attributed to the inflammatory state of the heart due to immune rejection.
7. In figure 4B, while the sections of CM group are cross-sections, the CM+PA are long axis sections.
8. The findings in Figure 5 (HUVEC) are not new.
9. In the figures, please indicate the number of animals in each experiment and statistical method.
10. The writing needs improvement. For example, it is unclear which type of animal model was used. The first time rat is mentioned is on page 6! This should be mentioned earlier including the abstract.
11. There is no quantitative assessment of cell retention and survival (i.g. PCR for human genes).
12. The Echo transducer is not optimal for small animal imaging.
13. The authors should address the study's limitations.

References

1. Ichim, T. E., F. Solano, F. Lara, J. P. Rodriguez, O. Cristea, B. Minev, F. Ramos, E. J. Woods, M. P. Murphy, D. T. Alexandrescu, A. N. Patel and N. H. Riordan (2010). "Combination stem cell therapy for heart failure." *Int Arch Med* 3(1): 5.

Response to Reviewers

We appreciate the detailed and insightful reviews of our manuscript. We addressed all of the issues that were raised to best of our ability and carried out additional experiments where necessary. All changes were incorporated into the manuscript accordingly. In the following Response to Reviewers, we provide a point-by-point response to each of the issues raised by the editors and reviewers. Each comment is addressed with reference made to the locations of any changes in the manuscript. The revised parts in the manuscript are highlighted in blue.

For this revision, we sought to employ a new experimental model which utilized hiPSC-CMs expressing green fluorescence protein (GFP) due to the mentioned limitations of CM-Dil (red fluorescence). While Dil is one of the most commonly used fluorescent dye for cell tracking experiments due to its strong signal intensity and long-lasting permanence, as the reviewers commented, Dil may not be exclusive to hiPSC-CMs within the heart tissues.

In light of these queries and to trace the signals from retained hiPSC-CMs within heart tissues more precisely, we have newly prepared hiPSC-CMs that continuously express GFP signal. We have prepared pMXs-eGFP retrovirus (**Figure A**) which was transduced into the cultured hiPSCs to first generate hiPSC-GFP (**Figure B-C**). Subsequently, hiPSC-GFP were differentiated into hiPSC-CMs-GFP through the previously described CM differentiation protocol (**Figure D and supplementary movie 6**). The detailed procedures for generating hiPSC-CMs-GFP are introduced in the revised methods section (page 19-20).

Several new *in vivo* experiments were performed using hiPSC-CMs-GFP. We induced myocardial infarction (MI) in rats by ligation of the left anterior descending coronary artery. Then we intramyocardially injected hiPSC-CMs-GFP to the MI-induced rat hearts in the presence or absence of hMSC-loaded patch (hMSC-PA) which was produced by using heart-derived decellularized extracellular matrix (hdECM). We harvested the rat heart tissues at various time points (2, 4, and 8 weeks) to investigate their phenotypic characteristics in the MI hearts.

By using these heart tissues, we have prepared the individual responses regarding reviewer's comments.

Generation of hiPSC-CM-GFP

(A) Structure of EGFP reporter system. (B) Representative images of hiPSC-GFP colony expressing GFP signal (C) Representative fluorescence-activated cell sorting (FACS) plots showing the percentage of GFP positive hiPSCs (hiPSC-GFP). These hiPSC-GFP were sorted out for subsequent expansion and further differentiation into the cardiomyocytes (D) Representative images of hiPSC-CMs-GFP derived from hiPSC-GFP.

Comments from Reviewer 1

1. BMC-MSCs are a widely used and well characterized stem cell type in cardiac injury, with known improvement of cardiac function via fusion with resident cardiomyocytes, transdifferentiation into myocytes or vasculatures. In this manuscript, the authors mainly focused on angiogenesis from MSC.

Have the authors examined if any patched MSC migrated into the myocardium? If so, what is the quantity of MSC detected and what lineage they adopted?

Response: We appreciate the reviewer 1 for raising this important point. According to the reviewer's suggestion, we investigated whether human mesenchymal stem cells (hMSCs) loaded in the patch (hMSC-PA) generated by using the heart-derived decellularized extracellular matrix (hdECM), migrated to the myocardium over the time.

To monitor migration of hMSCs, we pre-labeled hMSCs with CM-Dil (red fluorescence) prior to patch generation. Subsequently, we intramyocardially injected the hiPSC-CMs-GFP and implanted the hMSC-PA to the MI induced rat hearts as performed previously. Rat heart tissues were harvested at 4 and 8 weeks post-implantation to examine hMSC migration.

Microscopic observations revealed that the majority of Dil positive hMSCs were detectable within the patch (n=3). Furthermore, the myocardium did not express Dil positive hMSCs as shown in the following figures.

Based on these observations, we conclude that the majority of hMSCs were located within patch until 8 weeks and the hiPSC-CMs we observed in the myocardium did not originate from hMSCs.

We have included these new results in the revised manuscript including Results (Page 7), and Supplementary Figure 3.

Have the authors checked how long can MSC survive inside the patch after attachment in vivo (eg. Examine the patch with cell viability markers, proliferation markers, or cell death markers)? Are there vasculature formed inside of the patch?

Response: The hMSCs appeared to survive within the patch until 8 weeks post-implantation.

In the TUNEL assay, we observed that less than 10% of the hMSCs expressed TUNEL positive signals and the majority expressed both Dil and CD90, a specific MSC marker. These results clearly indicate that the majority of hMSCs within the patch remained viable even after 8 weeks.

In addition, Immunohistochemistry with CD31 antibody showed that there were no vessel like structures within the patch (Data not shown).

We have included these new results in the revised manuscript including Results (Page 7), and Supplementary Figure 4.

2. For a direct combinatory approach, have the authors tried directly combining iPSC-CM and MSC in porcine ECM, and inject this dual cell + ECM mixture for MI treatment? And what is the comparison between directly combination vs. using the patch?

Response: Reviewer 1 has raised an excellent query. As the reviewer suggested, we newly explored the therapeutic potential of a dual injection group (intramyocardially injecting a mixture of both hiPSC-CM and MSC together with the hdECM).

We newly performed the *in vivo* experiments with following experimental groups: **1)** MI control, **2)** Injection group with cell mixture (Dual injection: Dual IN): Injected 1×10^6 hiPSC-CMs, and 1×10^6 hMSCs together with the hdECM, and **3)** Complementary group with the hMSC-patch (CM + PA): Injected 1×10^6 hiPSC-CMs and implanted the hMSC-loaded patch made by using hdECM. Six rats per group were tested. To examine their therapeutic effects, echocardiography was performed at 2, 4 and 8 weeks after intervention (IN).

Our echocardiography results showed that both EF and FS of the complementary group was significantly higher than the mixed group at 4 and 8 weeks (**Supplementary figure 6**). These results indicate that therapeutic potential of Dual IN group was lesser than that of CM + PA group.

We included these new results as Supplementary Figure 6 in the Results section (Page 8-9).

3 & 4. One of the major claims of this manuscript is "dramatic increase in retention". However, no quantification was provided in in vivo iPSC-CM delivery in order to support this claim.

Of 1 million hiPCS-CM delivered, how many CM (with striation morphology) were detected at 2, 4, 8 weeks? Are these cells exclusively from iPSC-CM or could it be migrated MSC from the patch?

In order to support increased viability, please provide cell death assessment in in vivo assays including co-immunostaining and quantification with apoptotic/TUNEL markers of iPSC-CM after injection.

Response: We thank the Reviewer for this critical comment. As suggested by the reviewer 1, to directly quantify the number of hiPSC-CMs that remained within heart tissues, we intramyocardially injected 1×10^6 hiPSC-CMs-GFP into the MI induced rat hearts with or without hMSC-PA and harvested the tissues at 2, 4, or 8 weeks after injection.

Since it was very difficult to pinpoint every single hiPSC-CM, we aimed to quantify three different locations of the heart tissues (**Figure 4C**). Briefly, we prepared the heart tissue sections at three different locations and then immuno-stained with myosin heavy chain

(MYH6/7) antibody, which can detect both isomers (α and β forms) of myosin heavy chain proteins (MYH6/7). Subsequently, we imaged entire left ventricle (LV) area at three different location of heart sections and manually counted the hiPSC-CMs positive for both GFP signal (green) and MYH6/7 antibody (red).

There was significant difference in the number of hiPSC-CMs between the two groups from 2 weeks. The CM + PA group displayed a substantially higher count than that of the CM only group (Figure 4C). These results clearly and consistently support the notion that hMSC-PA enhance the retention of intramyocardially injected hiPSC-CMs.

These results clearly and consistently support the notion that hMSC-PA enhance the retention of intramyocardially injected hiPSC-CMs.

Of note, the morphology of hiPSC-CMs in the CM group exhibited a typically immature globular phenotype, while the CM + PA group exhibited a much larger rod-shaped structure that resembled adult-like CMs.

In addition, as suggested by reviewer 1, to confirm the survival of the hiPSC-CMs remained in the MI induced hearts, we performed TUNEL staining by using the heart tissues from the CM only or CM + PA groups harvested 8 weeks post intervention. We counted the number of cells that overlapped with the GFP (green), MYH6/7 (red) and the TUNEL signals (grey).

As shown in the figure, only about 10% of the hiPSC-CM-GFP were positive for TUNEL signal in both experimental groups. These results suggest that the majority of intramyocardially injected hiPSC-CMs in the presence or absence of hMSC-PA remained alive in the MI hearts for at least 8 weeks (Figure 4D).

We have included these new results in the revised manuscript, including Methods (Page 19-20), Results (Page 10), and Figure 4C-D.

5. Dil was used as a cell tracker in intramyocardial injection. Please provide the Dil dilution used methods.

Dil is a cell membrane free entry sphere, which does not reflect the host cell morphology. Its strong intensity can shine through neighboring tissue, therefore can make cell source interpretation very confusing with the resident tissue which are not derived from the injected iPSC-CM. Dil cannot be used as the exclusive marker for viable cells. Although authors provided hTNNI3 staining in Fig. 4C, the image quality is too poor to be analyzed. The hTNNI3 pattern does not resemble cardiomyocyte striation as seen in Fig. 4B (CM+PA panels).

Please provide better quality images of co-immunostaining of human cardiomyocyte marker or human species marker (HNA, hMito) in addition to Dil to exclusively confirm cell source (Fig 4).

Response: We thank the reviewer for this excellent comment. In the previously submitted study, we used CM-Dil (Chloromethylbenzamido, CellTracker[™]) to facilitate the tracking of those intramyocardially injected hiPSC-CMs within the heart tissues. The stock solution of CM-Dil was prepared using a modification of the manufacturer's instructions. From a 1 mg/ml CM-Dil stock solution in DMSO, 5µM solutions were made in 500µl of DMEM and this working solution was labelled the hiPSC-CMs. Indeed, Dil has been widely used for the cell tracking

purpose due to its powerful and long-lasting signal. However, as suggested by reviewer 1, mostly because of its strong signal intensity, it can be seen with other surrounding cells and it may not be able to ensure the specificity of their signals. In this regard, in our revised study, we employed hiPSC-CM-GFP showing GFP signal all the time to be able to track the hiPSC-CMs more accurately.

To verify the identity of hiPSC-CM-GFP in the heart tissues, we performed immunohistochemistry with antibodies targeting human beta myosin heavy chain protein (hMYH7; grey) as well as human mitochondria (hMito; grey) using the rat heart tissues at 8 weeks. Confocal microscopy images clearly confirmed significant expression of either hMito or hMYH7 cells in the hiPSC-CM-GFP found in the heart tissues. While the antibody targeting both MYH6/7 (red) or TNNI3 (red) proteins was detected in both rat CMs and hiPSC-CM-GFP, hMito or hMYH7 were only seen in hiPSC-CM-GFP indicating that the GFP signals were from hiPSC-CMs.

We have included these new results in the revised manuscript which are now presented in the Methods (Page 20), Results (Page 10), and **Figure 4A & B**.

6. It is interesting to see tight junction of iPSC-CM to resident CM. However, the area chosen in Fig 4B are not representative, both CM groups are resident myocyte sparse, whereas both CM+PA groups are with nicely aligned cardiomyocyte.

Please provide images of comparable and representative regions across CM and CM+PA groups. In addition, in order to exclusively confirm human and rat myocyte share tight junction, please provide higher magnification images of GJA1 co-staining with human cardiomyocyte marker and rat myocyte marker on each side of GJA1.

Response: Reviewer 1 has raised another excellent point. As a matter of fact, the representative images of both experimental groups showed in **Figure 4B** in the previously submitted manuscript, were among the best images we could take. In the previously submitted results, we continuously observed that the hiPSC-CMs in the CM only group were much smaller in size and their sarcomere structure was not well developed as an immature phenotype. On the other hand, the size of the hiPSC-CMs in the CM+PA group was significantly larger in size, and the myofibril structure was organized much better. More importantly, hiPSC-CMs in the CM+PA group were well incorporated with the host rat CMs.

Since the morphologies between two groups were excessively different, we paid particular attention to the hiPSC-CMs in the CM only group and we have repeatedly examined to make sure that what we have seen was correct. We made significant efforts to find better images of hiPSC-CMs in the CM only group. However, the images we originally submitted were the best images we could obtain.

In this newly prepared study with hiPSC-CM-GFPs, we observed the similar result; the size of GFP-positive hiPSC-CMs in the CM only group was still smaller and appeared to be immature CM phenotype. In contrast, hiPSC-CM-GFP in the CM+PA group consistently displayed as a rectangular shape which is typical shape of mature CMs (**Figure 4**).

More importantly, similar with previous results, we could easily detect that many of hiPSC-CMs in the CM + PA group formed gap junctions with host rat CMs through abundant expression of the connexin 43 (GJA1). Whereas, we concluded that hiPSC-CMs in the CM only group did not form the gap junction with the host CMs. These results strongly suggest that secreted factors from hMSC-loaded patch improved the maturity of hiPSC-CMs in the MI hearts. We have included these new results in the revised manuscript in the Results (Page 10-11) as Figure 4F.

We appreciate the reviewer for this great input.

7. In all microscopy images, please provide co-staining with nuclear counterstaining, background tissue staining (eg. myocardium, cytoskeleton, cell boundaries) in order to provide reference landmark for interpretation.

Response: We thank the reviewer's comment. As suggested by reviewer 1, we counterstained all newly submitted images with designated antibodies. In particular, all our images obtained from cardiac tissue were counterstained with CM specific antibodies such as myosin heavy chain 6/7 (MYH6/7), cardiac troponin T (TNNT2) or cardiac troponin I (TNNI3).

8. The hiPSC-CM contraction seems unsynced. Please provide physiological data to show if hiPSC-CM xenograft caused arrhythmia to host rat heart.

Response: We thank the Reviewer for this important comment. As reviewer suggested, to investigate whether hiPSC-CMs could form the physiological synchronization with rat host CMs and to examine their arrhythmogenic potential, we performed multi-electrode arrays (MEAs) analyses using the co-culture model of hiPSC-CMs and neonatal rat ventricular CMs (NRVM).

For MEA analyses, we plated the hiPSC-CM-GFP and NRVM on the MEA chambers and analyzed their electrical signals (**Supplementary figure 14**). Positively, within 48 hours of co-culture, we were able to detect synchronous mechanical activity between hiPSC-CM-GFP and NRVM (**Supplementary movie 7**).

Next, we utilized the MEA mapping technique to evaluate the functional interactions within the co-cultured CMs. By recording extracellular field potentials simultaneously from all electrodes, we were able to create activation maps showing the development of synchronized activity in all hybrid cultures. Co-cultured hiPSC-CM-GFP and NRVM displayed well synchronized action potential propagation and did not show arrhythmogenic beating during 20 minutes of recording (**Figure B and supplementary movie 7**).

Analysis of electrical activity between hiPSC-CMs and neonatal rat ventricular cardiomyocytes through multielectrode arrays.

(A) Field image of co-cultured neonatal rat ventricular cardiomyocytes (NRVM) and hiPSC-CMs on MEA chip. **(B)** Isochronal maps of spontaneous AP propagation recorded from co-cultured NRVM and hiPSC-CM for 200 s. The activation time is represented at the right of the map. **(C)** Conduction velocity of NRVM, hiPSC-CM, and co-cultured NRVM/hiPSC-CM. **(D)** A representative MEA extracellular recording from co-cultured NRVM/hiPSC-CM. **(E)** Temporal development of conduction velocity from NRVM (black circle), hiPSC-CMs (red circle), and NRVM + hiPSC-CMs (blue circle)

Of note, the conduction velocity of co-cultured CMs was intermediate level between hiPSC-CM-GFP and NRCMs. As the above figure showed, the conduction velocity of co-cultured CMs was lower than NRVM only but higher than hiPSC-CMs only (**Supplementary figure 14C & E**). We carefully conclude that the conduction speed of NRVM was reduced by co-cultured hiPSC-CMs. Nevertheless, the results of intermediate conduction speed and uniformed propagation are likely to be evidences of synchronized contraction between hiPSC-CM-GFP and NRVM.

In summary, these new results clearly suggest that hiPSC-CMs displayed synchronous electrical activities with NRCMs and did not show any evidence of arrhythmia.

These new data were included in Methods (Page 24), Results (Page 11) and placed in the **Supplementary Figure 14** and the **supplementary movie 7**.

9. In all cardiac function measurements, the basal level (pre) of EF (~40%) or FS (<20%) seemed very low as a healthy normal heart.

At post-MI time points, the drop of EF or FS is within ~10% over 8 weeks even in control (no intervention) group. A 10% difference in EF is very minimal as an infarct damage. In fact, the CM+PA kept the EF and FS constant throughout the post-MI time course with no functional compromise at all.

This function data is inconsistent with fibrosis staining where 80% of tissue are fibrotic in control animal, but functionally only lost 10% EF (Fig 2B, 3A, 3C).

Can the authors comment on this inconsistency? In order to rule out technical inconsistency, please provide representative echo image/video, long/short axis, reference heart rate (RPM), and body temperature data. In all in vivo assays, please provide sham throughout all time points to set a comparison reference.

Response: We apologize for any confusion caused by unclear wording. In fact, EF and FS, which are shown as “PRE” in the X axis in the figures, were **NOT** the functional measurements of a normal control group as a basal level, but are the measurements obtained from the heart one week after MI induction.

As shown in the methods section, we typically induced MI by permanent ligation of LAD and measured cardiac function by echocardiography after 1 week from the MI induction to ensure

that MI was successfully induced. Only for those rats whose functional parameters were met [EF (< 45%) or FS (< 20%)], we applied the intervention (IN) such as intramyocardial injection of hiPSC-CMs and/or implantation of hMSC-patches.

As reviewer 1 recommended, we newly generated the Sham model of rats (n=6) and included their echocardiography data throughout all time points (2, 4, and 8 weeks) to set as a comparison reference. Compared with the sham group, MI-induced rats displayed the reduction of the EF and FS by more than 40% and 20%, respectively. We consider that about 40% reduction in EF is appropriate as an infarct damage and consistent with fibrosis staining where 80% of tissue were fibrotic in MI control hearts. To prevent further confusion, we revised the **Figure 2B and 3A**. In these new figures, we changed the terminology of **PRE, 2W, 4W, and 8W** to **PRE-intervention (PRE-IN), 2W-IN, 4W-IN, and 8W-IN**, respectively. In addition, according to reviewer's comment, we newly incorporated the sham echo data into the **Figure 2B and 3A**.

We also prepared the representative echo images, long and short axis movies and the reference heart rate (RPM) in all experimental groups. These new data were included in Results (Page 8-9), Figure 2B, 3A, Supplementary Figure 8 and the supplementary movie 4 & 5.

10. PA only group is missing in Fig 3. Although it was shown in Fig 2, it'll be constructive to see all four groups quantified and graphed together for direct comparison.

Response: We thank the Reviewer for this suggestion. As suggested by reviewer 1, the revised Figure 3A by combining all four experimental groups for easier direct comparison.

11. According to figure legend, Fig 2A shows representative area of capillaries in border zone, however the quantification is from infarct zone.

In order to claim the increased density of capillaries are exclusively benefit from PA, please provide images and quantification for both border zone and infarct zone in comparison to remote zone (area that has no direct contact with PA). It'll also be helpful if authors can label (with dash lines) bz/iz on images as they did in Fig 4A.

Response: We thank Reviewer 1 for this excellent comment and apologize our mistake. As suggested by reviewer 1, we newly took the images of capillaries from all three areas including border zone, infarct zone and remote zone and quantified the number of capillaries (IsB4; green) in all three area. Below figures are the representative images of capillaries from all three areas and their quantification summary. The results show that although the number of capillaries in the remote zone in all three groups were not differ significantly, the number of capillaries in the border Zone and infarct zone in the CM + PA group was significantly greater than two other groups.

We included the new data in the previous Figure 2A and Figure 3B. and described the results on page 6-7 and page 9.

MYH6/7 IsB4 DAPI

12. In Discussion, authors claimed that the patches can prevent expulsion of injected hiPSC-CM into epicardial space. The suturing of patch seems an invasive procedure to the epicardium. As from the video provided, it seems patch stayed outside of epicardium, which is perhaps ruptured or punctured by suturing. Please specify if the patch is engineered inside or outside of the epicardium?

Response: We apologize for any confusion we raised regarding the patch implantation. Indeed, the patches were implanted to the outside of the epicardium as we described in the method section. Due to excellent adhesive property of hdECM, the hMSCs-PA were well attached to the epicardium without any additional treatment. However, to ensure the patches are well attached to the epicardium throughout the experiments, we sutured them twice. There was no severe bleeding after sutures and no significant evidences for rupture. In addition, as shown in the figures in the response 1, the patches were placed in the epicardium up to 8 weeks from implantations. We have added an additional description regarding the location of patch in the method section.

13. The authors claim that the hiPSC-CM are 98% purity (TNNI3). However, it is surprising to see beating cardiomyocytes migrate towards MSC (sup Fig 5). Why would beating myocytes migrate? Are there any MSC migrate towards myocytes? What is the benefit of a migrating cardiomyocyte?

In order to support the claim, please provide migrating assay with fluorescently labelled hiPSC-CM and MSC to separate these two cell types with two colors, or provide live imaging video.

Response: We thank the Reviewer for this helpful comment. In fact, we were also found interests that hiPSC-CMs were migrated to the hMSCs. In order to address the reviewer's comment, we performed the CM migration assay again with hiPSC-CM-GFP and hMSCs. As negative control groups, we also included hiPSCs culture alone or in the co-culture with HUVEC.

Briefly, hiPSC-CM-GFP and hMSCs or HUVEC were cultured on Culture-Insert 2 well (ibidi) and culture inserts were removed after 24 hours to monitor migration of hiPSC-CM-GFP, space between cells was digitally photographed with 3 minutes intervals for 72 hours. As results, we observed that hiPSC-CM-GFP began to migrate towards hMSCs after 12 hours and reached the center of the dish by 48 hours. Within 72 hours, hiPSC-CM-GFP (white arrow) continued to approach periphery of hMSCs which is indicative of cell-cell affinity. However, hiPSC-CM did not migrate significantly when hMSC was absent or when hiPSC-CMs were co-cultured with HUVEC. Of interest, as Supplementary movie 8,9, and 10 showed, hMSCs also migrated towards hiPSC-CMs but its speed is much slower than migration pace of hiPSC-CMs.

Based on the results obtained from our experiments and the finding from others, we carefully concluded that hiPSC-CMs migrate to the hMSCs due to the paracrine factors released from hMSCs as chemokinetic responses. Indeed, hPSC-CMs have been reported to show robust promigratory behavior to ECM components and soluble signaling molecules such as Fibronectin and Wnt5a, respectively. We consider that the capability of CMs to migrate to

damaged areas is prerequisite for repairing damaged area in hearts *in vivo* and therefore beneficial, as evidenced in adult zebra fish which can regenerate heart tissue through the migration and proliferation of pre-existing CMs. These new data were included in Results (Page 12-13) and placed in the Supplementary Figure 15 and Supplementary Movie 8,9, and 10

Migration of hiPSC-CMs towards hMSCs.

(A) Schematic representation of monolayer co-culture system to assess the migration of hiPSC-CM to the hMSCs. **(B)** Representative images of hiPSC-CM-GFP culture alone or co-cultures of hiPSC-CM-GFP with HUVEC or hMSCs. Red box and white arrows indicate the migration area at specific time points and the hiPSC-CM-GFP.

Minor comments

1. Cardiac patch has been reported before with various cell types (endothelial cells, smooth muscle cells, iPSCs, cardiomyocytes, etc) and binding material (collagen, fibrin, hydrogel, etc). Please cite these publications in your introduction.

2. This study is not "the first to simultaneously examine the effects of two distinct major stem cell types" in cardiac repair. Multiple previous studies reported directly injection of combinatory cell therapy, eg. MSC+CPC (Williams et al, Circulation, 2013), (Quijada et al, Circulation Research, 2015). Please cite these references.

Response: The authors are grateful for these important comments. As recommended by the reviewer we have revised the discussion and cited the suggested papers (page 14).

3. It is interesting to see an increased number of capillaries in PA engineered heart. In order to make sure these capillaries are properly functional, please also provide quantification of diameter ranges (Fig 2A, Fig 3B).

Response: Thanks for the advice from reviewer 1. As reviewer suggested, we newly measured the diameter of all capillaries and generated a new figure summarizing these results.

As figure showed the number of capillaries with a diameter range of 5 ~10 mm in the both infarct zone and border zone in the hearts from the dually treated group were substantially higher than other control or hiPSC-CMs only groups. These new data were included in Results (Page 9) and placed in the Supplementary Figure 7.

4. Please keep experimental procedure to method section. Some content in result section should belong to methods.

Response: Authors are grateful for this helpful comment. As reviewer 1 suggested, we have removed redundant description of experimental methods in the results section.

5. Please provide size of scale bar.

Response: As reviewer recommended, we placed the size of scale bar within all the figures.

6. What is the purpose of placing a PE tubing into LAD? And was the tubing ever removed?

Response: The purpose of placing a PE tubing into LAD is to identify that MI is induced properly. After LAD ligation, we generally check the color of heart to confirm whether MI is induced correctly. If MI is not induced properly, we untie the LAD and ligate again. The tubing is helpful to untie the LAD. Once we confirm that MI is induced correctly, we immediately remove the tube from LAD.

7. Please specify the injection volume for 1 million cells.

Response: As reviewer 1 suggested, we have specified the number of injected hiPSC-CMs (1×10^6) in the revised method section (page 20).

8. Discussion session is redundant data presentation and too much speculation.

Response: We thank the review for point these out. We have revised the discussion as reviewer 1 recommended.

9. Reference 19 and 40 are redundant.

Response: We thank the review for pointing this out and apologize our mistake. We have removed the redundant reference in the reference section.

Comments from Reviewer 2

1) *The most significant progress in this manuscript, in my view, would be cardiac graft maturation by co-transplantation of hMSC-PAs. Grafted CM shown in Fig. 4 and Supplementary Fig. 4 look very matured CMs which is surprising to me. Are these really graft derived human cells?*

In fact, these cells strongly expressed MYH6, commonly expressed rodent CMs but not in human. I would like to know if these CM are unequivocally graft-derived cells by following experiments.

a) Please check the expression of MYH7 in graft CMs. In our hands, host (rat) CMs almost exclusively express MYH6 and graft (human) CMs express only MYH7.

The authors should try human specific antibody with chromogenic staining to avoid autofluorescence.

Response: We appreciate the Reviewer 2 for this excellent comment. In fact, reviewer 1 also provided similar comment regarding the identity of hiPSC-CMs observed in the heart tissues. Please refer to our response to the reviewer 1's comment 5.

To verify the identity of hiPSC-CM-GFP in the heart tissues, we performed immunohistochemistry with antibodies targeting human beta myosin heavy chain protein (hMYH7; grey) as well as human mitochondria (hMito; grey) using the rat heart tissues at 8 weeks. Confocal microscopy images clearly confirmed significant expression of either hMYH7 or hMito in the hiPSC-CM-GFP found in the heart tissues. While the antibody targeting both MYH6/7 (red) or TNNI3 (red) proteins was detected in both rat CMs and hiPSC-CM-GFP, hMYH7 or hMito was only seen in hiPSC-CM-GFP indicating that the GFP signals were from hiPSC-CMs.

We have included these new results in the revised manuscript which are now presented in the Methods (Page 20), Results (Page 10), and **Figure 4A & B**.

b No attempts have been made to elucidate the mechanisms of cardiac maturation.

Likewise experiments in Fig .5, in vitro maturation study would be required.

Response: We thank the reviewer 2 for this critical comment. As reviewers 2 recommended, we conducted a series of additional *in vitro* experiments to verify the effect of hMSCs on CM maturation shown in vivo.

Briefly, we collected the conditioned media (hMSC-CM) from the hMSC cultures (passage 4) and treated different percentage of hMSC-CA to the freshly isolated neonatal rat ventricular cardiomyocytes (NRVM). With these NRVM received hMSC-CM, we performed qRT-PCR analyses, immunocytochemistry and CCK-8 assay to examine its effects on CM maturation.

At first, qRT-PCR results demonstrated that treatment with hMSC-CM into the NRVM did not significantly alter the expression level of genes related to CM maturation (Data not shown). However, the results from cardiomyocyte area measurement and the Cell Counting Kit-8 (CCK-8) array demonstrated that 10 or 30% of hMSC-CM dramatically increased the size and the number of NRVM.

Indeed, hypertrophy and is considered as one of the important parameters for CM maturation as it is critical part of normal heart development. The size of the hESC-CMs has been reported to be approximately 600 μm^2 , which is significantly smaller than an adult CM, Since cell size greatly influences impulse propagation, maximal rate of action potential depolarization and total contractile force, thus cell size is an important parameter for determining CM maturation.

Several previous studies demonstrated a number of approaches for inducing CM maturation including prolonged culture (up to 100 days), cyclic stress, and adrenergic agonists markedly increased CM size as a hypertrophy, and proliferation rates were moderately upregulated.

In summary, based on these results, we confirm that the factor(s) secreted from hMSCs contributes the maturation of CMs at least to a certain extent.

We have included these new results in the revised manuscript, including Results (Page 10), and Supplementary figure 12.

2) The authors claimed that hMSC-PAs secreted paracrine factors by RT-PCR of hMSC-PA transplanted and non-treated hearts. I feel cell-free patch transplanted control is required.

Response: We thank the reviewer 2 for this insightful comment. As reviewer 2 suggested, we newly generated cell-free patch and implanted MI induced rat hearts. 1 week after implantation, we harvested the heart tissues and perform the qRT-PCR analyses. The results showed that expression levels of all previously examined genes from the cell-free patch implanted hearts were significantly lower than dual approach group. Their expression level

was similar to the MI group control hearts group. These results further verify that shown paracrine effects is derived from the hMSCs not from the patch.

These new data were included in Results (Page 6) and incorporated into the Figure 1E-G.

3) Important information is missing regarding time points of histology in Fig. 4.

Response: As reviewer 2 suggested, we have added a description of the time points in the figure 4 legend.

4) Control study is missing regarding migration study (Supplementary Fig. 5).

Response: We thank the Reviewer for this helpful comment. In order to address the reviewer's comment, we performed the CM migration assay again with hiPSC-CM-GFP and hMSCs. As negative control groups, we also included hiPSCs culture alone or co-culture with HUVEC.

Briefly, hiPSC-CM-GFP and hMSCs or HUVEC were cultured on Culture-Insert 2 well (ibidi) and culture inserts were removed after 24 hours to monitor migration of hiPSC-CM-GFP, space between cells was digitally photographed with 3 minutes intervals for 72 hours.

As results, we observed that hiPSC-CM-GFP began to migrate towards hMSCs after 12 hours and reached the center of the dish by 48 hours. Within 72 hours, hiPSC-CM-GFP (white arrow) continued to approach periphery of hMSCs which is indicative of cell-cell affinity. However, hiPSC-CM did not migrate significantly when hMSC was absent or when hiPSC-CMs were co-cultured with HUVEC. Based on the results obtained from our experiments and the finding from others, we carefully concluded that hiPSC-CMs migrate to the hMSCs due to the paracrine factors released from hMSCs as chemokinetic responses.

Migration of hiPSC-CMs towards hMSCs.

(A) Schematic representation of monolayer co-culture system to assess the migration of hiPSC-CM to the hMSCs. (B) Representative images of hiPSC-CM-GFP culture alone or co-

cultures of hiPSC-CM-GFP with HUVEC or hMSCs. Red box and white arrows indicate the migration area at specific time points and the hiPSC-CM-GFP.

These new data were included in Results (Page 12-13) and placed in the **Supplementary Figure 15 and Supplementary Movie 8,9, and 10**

5) Some figure numbers in the text do not correspond to the actual figures (e.g. P. 6, line 5).

Response: We apologize for our mistake. We have revised the manuscript as the reviewer pointed out.

Comments from Reviewer 3

1 & 2. The concept is of great interest to the field of cell therapy. However, the idea is not entirely new. (Ichim, Solano, et al. 2010)

Some of the major findings in this study are not new as indicated.

Response: Authors appreciate that reviewer 3 found interest to our study. We agree that the idea of combinatory stem cell therapy is not completely new. However, most previous reports examining combinatory cell therapy were simply combining distinct types of stem cells together to enhance their intercellular interaction. Compared to previous studies, our study sought a different way to combine two different cell types through two different routes of cell delivery; intramyocardially injected hiPSC-CMS and an epicardially implanted hMSCs-loaded patch (hMSC-PA) produced by using the heart-derived decellularized extracellular matrix (hdECM) to provide structural and potential vasculature support to the MI hearts.

In addition, in regard to the cardiac patch, although cardiac patch has been reported before for cardiac repair, we intended to use hMSC-PA to enhance vascular regeneration through prolonged secretion of angiogenic paracrine factors in MI hearts. In addition, our study employed innovative engineering approach using lyophilized hdECM and hMSCs as bioink on a 3D printed scaffold. Lyophilization of hdECM allows convenient storage and transportation of cell-free ECM. Therefore, our strategy potentially provides translational and clinical benefits to the field of cardiac repair.

3. The authors used human iPSC-CM and human MSCs loaded on a patch generated from a porcine heart in a rat model of MI. The risk of xenogeneic cells rejection is significant.

Response: Reviewer has raised an excellent query. We accept that intramyocardial injection of hiPSC-CMs and implantation of hMSC-PA into the rats can induce xenogeneic rejection. Thus, to prevent immune rejection, we used following immunosuppressants; azathioprine, 2mg/kg, cyclosporine A, 5 mg/kg, and methylprednisolone, 5mg/kg per daily as we previously reported. By treating these immunosuppressants, we did not observe a significant evidence for immune rejection throughout the experiments. Indeed, our current study is a proof of concept approach hoping to ultimately move forward to human clinical application. Thus, we expect that xenogeneic rejection will not occur if our strategy with hiPSC-CMs and hMSCs is applied to human patients.

4. The authors did not evaluate the effect of treatment on cardiac remodeling and the morphometric assessment in figure 3C is minimal.

Response: Effects of cardiac remodeling thought all experimental groups were evaluated by both cardiac echocardiography measuring left ventricular internal diameter end diastole (LVIDd) and end systole (LVIDs), Septal wall thickness, Posterior wall thickness, Relative Wall Thickness. We also performed Masson's Trichrome (MT) staining to determine cardiac fibrosis. While we agree with the reviewer's comment that morphometric assessment may provide the

limited information on cardiac remodeling, we respectfully claim that both morphometric assessments are still one of the most widely used methods for measuring cardiac remodeling.

5. The authors should add characterization of the hMSC-PA in vivo (during MI)- for example, cell survival in the patch, histological staining of the patch.

Response: In fact, reviewer 1 also provided similar comment regarding the identity of hiPSC-CMs within the hearts. Please refer to our response to the comment 5 from the reviewer 1.

We found that the vast majority of hMSCs appear to be well survived in the patch. In the TUNNEL assay using the 8 weeks post MI rat heart tissues, only less than 10% of the hMSCs expressing appeared to be TUNNEL positive cells. Most hMSCs expressing CD90 surface protein, a specific marker for hMSC, were not positive for TUNEL.

These results clearly suggest that the majority of hMSCs were remained alive within the patch even after 8 weeks from the patch implantation.

We have included these new results in the revised manuscript including Results (Page 7), and Supplementary Figure 4.

6. Statistic: changes along time should be tested with repeated measure ANOVA.

Response: We thank the Reviewer's comment regarding statistics. However, we respectfully disagree with Reviewer's comments that repeated measure ANOVA is required.

The choice of one-way ANOVA as a multiple comparison procedure was considered appropriate because it is an easily interpretable test that fits what our data show (bar graph rather than linear curve over times). More importantly, since our analyses was restricted to one time point (2, 4, or 8 week) rather than multiple time points together over time, a one-way ANOVA was deemed to be appropriate. It is important to note that each group is subjected to at most one intervention (independent variable). The analyses undertaken have one dependent variable analyzed at a time. MI is induced in all groups and results in identical ejection fraction (EF) or fractional shortening (FS) in all groups.

7. In figure 2A and 3B, the authors show a higher number of capillaries in the infarct zone of MSC-PAs and MSC-PAs+ iPSC-CM implanted hearts compared with control. This could be attributed to the inflammatory state of the heart due to immune rejection.

Response: We respectively disagree with the comment that shown higher number of capillaries in the hMSC-loaded patch (PA) only or hiPSC-CMs with PA (CM + PA) group could be due to immune rejection. Indeed, immune rejection through immune suppressants were applied to all experimental groups including MI control and hiPSC-CMs only injected groups which displayed significant lower number of capillaries compared to PA or CM + PA group.

8. In figure 4B, while the sections of CM group are cross-sections, the CM+PA are long axis sections.

Response: We apologize that we created confusion for the reviewer. Indeed, all images used in our study were taken from cross-sectioned heart tissues. The picture shown in figure 4B from both experimental groups were also taken from cross-sectioned heart tissues. Those photos were just taken in different directions.

9. The findings in Figure 5 (HUVEC) are not new.

Response: We agree that the results of Matrigel plug assay by using HUVECs designed to examine the effects of hMSC-conditioned media *in vitro* were not novel. We just intended to confirm what we observed from *in vivo* studies through *in vitro* Matrigel plug assay with HUVECs since this is one of widely used *in vitro* models for testing angiogenic effects.

10 & 11. In the figures, please indicate the number of animals in each experiment and statistical method.

The writing needs improvement. For example, it is unclear which type of animal model was used. The first time rat is mention is on page 6! This should mention earlier including the abstract.

Response: We thank the reviewer 3 for pointing these out. As the reviewer recommended we have included the number of animals and statistical method in the figure legends. In addition, we have specified animal model throughout the revised manuscript including abstract.

12. There is no quantitative assessment of cell retention and survival (i.g. PCR for human genes).

Response: The authors thank the reviewer for this insightful comment.

Reviewer 1 also provided similar comment regarding the identity of hiPSC-CMs within the heart tissues. Please see to our response to the reviewer 1's comment 5.

To directly quantify the number of hiPSC-CMs remained within the heart tissues, we intramyocardially injected 1×10^6 hiPSC-CMs-GFP into the MI induced rat hearts with (CM + PA)/without (CM only) hMSC-loaded patch and harvested the heart tissues at 2, 4, or 8 weeks after injection.

Since it was very difficult to pinpoint every single hiPSC-CM, we aimed to quantify three different locations of the heart tissues (**Figure 4C**). Briefly, we prepared the heart tissue sections at three different locations and then immuno-stained with myosin heavy chain (MYH6/7) antibody, which can detect both isomers (α and β forms) of myosin heavy chain proteins (MYH6/7). Subsequently, we imaged entire left ventricle (LV) area at three different location of heart sections and manually counted the hiPSC-CMs positive for both GFP signal (green) and MYH6/7 antibody (red).

There was significant difference in the number of hiPSC-CMs between the two groups from 2 weeks. The CM + PA group displayed a substantially higher count than that of the CM only group (**Figure 4C**). These results clearly and consistently support the notion that hMSC-PA enhance the retention of intramyocardially injected hiPSC-CMs.

These results clearly and consistently support the notion that hMSC-PA enhance the retention of intramyocardially injected hiPSC-CMs.

Of note, the morphology of hiPSC-CMs in the CM group exhibited a typically immature globular phenotype, while the CM + PA group exhibited a much larger rod-shaped structure that resembled adult-like CMs.

In addition, as suggested by reviewer 1, to confirm the survival of the hiPSC-CMs remained in the MI induced hearts, we performed TUNEL staining by using the heart tissues from the CM only or CM + PA groups harvested 8 weeks post intervention. We counted the number of cells that overlapped with the GFP (green), MYH6/7 (red) and the TUNEL signals (grey).

As shown in the figure, only about 10% of the hiPSC-CM-GFP were positive for TUNEL signal in both experimental groups. These results suggest that the majority of intramyocardially injected hiPSC-CMs in the presence or absence of hMSC-PA remained alive in the MI hearts for at least 8 weeks.

We have included these new results in the revised manuscript, including Methods (Page 19-20), Results (Page 10), and Figure 4C-D

13. The Echo transducer is not optimal for small animal imaging.

Response: The ultrasound used in our study is equipped with a 15MHz L15-7io linear transducer (Affniti 50G, Philips). This transducer has an operating extended frequency range of 7-15 MHz. Since the optimal frequency range for the rat echocardiogram is approximately 12 MHz (12 ~ 14 MHz,) thus our echo transducer is sufficiently capable of rat heart imaging. We also agree that there are other better methods for small animal imaging including MRI. However, we respectfully suggest that Echo transducer is still one of the most widely used methods for small animal imaging.

14. The authors should address the study's limitations.

Response: We appreciate Reviewer 3's comment. As reviewer 4 recommended, we added following study's limitations in Discussion in the revised manuscript on page 16.

Limitations.

Even though this study evidenced several promising results, there are some limitations to the study that require consideration. Since our complementary approach (CM + PA) was examined in an acute ischemia model, the outcomes may be different if applied to models with advanced heart failure. In addition, this approach necessitates somewhat complicated and technical surgical procedures to successfully inject hiPSC-CMs and implant hMSC-PA to the heart. Follow-up studies should be carried out to develop a more concise surgical method that can take full advantage of this approach with less invasive procedures.

Reviewer #1 (Remarks to the Author):

Authors have provided intensive amount of data in the revision, they've done an amazing job to well address my original concerns. All questions are well-explained, very impressive. I have no further comments.

Reviewer #2 (Remarks to the Author):

The authors worked to respond reviewers' comments raised in the first round of review. I found the manuscript improved significantly. I have no further comments.

Reviewer #3 (Remarks to the Author):

The authors made many improvements. However, the study still suffers from significant limitations:

1. The subject and findings lack novelty.
2. The method of heart imaging (clinical system with 15MHz transducer) is outdated.
3. I disagree with the persistent use of T-test. The authors should consult a statistical advisor. Changes along time such as heart function (Figures 2, 3, 5) should be tested by 2- repeated-measures ANOVA with appropriate post hoc testing. By using a simple T-test for each time point, the authors artificially increased their chance to find statistical significance. The abuse of simple T-test and wrong statistical analyses are one of the primary reasons for irreproducibility in the present field. The authors should read classical papers in the field in leading journals (Caspi, Huber et al. 2007, Tiburcy, Hudson et al. 2017) or a recent position paper (Lindsey, Gray et al. 2018).

References

- Caspi, O., I. Huber, I. Kehat, M. Habib, G. Arbel, A. Gepstein, L. Yankelson, D. Aronson, R. Beyar and L. Gepstein (2007). "Transplantation of human embryonic stem cell-derived cardiomyocytes improves myocardial performance in infarcted rat hearts." *J Am Coll Cardiol* 50(19): 1884-1893.
- Lindsey, M. L., G. A. Gray, S. K. Wood and D. Curran-Everett (2018). "Statistical considerations in reporting cardiovascular research." *Am J Physiol Heart Circ Physiol* 315(2): H303-H313.
- Tiburcy, M., J. E. Hudson, P. Balfanz, S. Schlick, T. Meyer, M. L. Chang Liao, E. Levent, F. Raad, S. Zeidler, E. Wingender, J. Riegler, M. Wang, J. D. Gold, I. Kehat, E. Wettwer, U. Ravens, P. Dierickx, L. W. van Laake, M. J. Goumans, S. Khadjeh, K. Toischer, G. Hasenfuss, L. A. Couture, A. Unger, W. A. Linke, T. Araki, B. Neel, G. Keller, L. Gepstein, J. C. Wu and W. H. Zimmermann (2017). "Defined Engineered Human Myocardium With Advanced Maturation for Applications in Heart Failure Modeling and Repair." *Circulation* 135(19): 1832-1847.

Response to Reviewers

We appreciate the insightful reviews of our manuscript. We addressed all of the issues that were raised to best of our ability. All changes were incorporated into the manuscript accordingly. The revised parts in the manuscript are highlighted in blue.

Comments from Reviewer 3

1. The subject and findings lack novelty.

Response: We regret that we had failed to convince Reviewer 3 regarding the novelty of our study. However, we respectfully claim that our study is considered to be novel compared to previous reports examining combinatory cell therapy which were simply combining distinct types of stem cells together to enhance their intercellular interaction. To the best of our knowledge, our study is the first to simultaneously examine the effects of two distinct major stem cell types delivered via two different routes for inducing comprehensive cardiac repair.

Our study sought a different way to combine two different cell types through two different routes of cell delivery; intramyocardially injected hiPSC-CMS and an epicardially implanted hMSCs-loaded patch (hMSC-PA) produced using the heart-derived decellularized extracellular matrix (hdECM) to provide structural and potential vasculature support to the myocardial infarction (MI) hearts.

As results, our dual approach collectively rejuvenated the myocardium and vessels post MI. Epicardially implanted hMSC-PA allowed a supportive microenvironment, which augmented vascular regeneration through prolonged secretion of beneficial paracrine factors. More importantly, hMSC-PA greatly improved the retention, distribution, engraftment, and maturation of intramyocardially injected hiPSC-CMs. Therefore, our strategy potentially provides translational and clinical benefits to the field of cardiac repair.

2. The method of heart imaging (clinical system with 15MHz transducer) is outdated.

Response: We appreciate Reviewer 3's comment. As the editor suggested, we added the following statement in the Discussion in the revised manuscript on page 16 as limitations to our heart imaging method.

Limitations.

In addition, the majority of cardiac imaging results was solely obtained by echocardiography in this study. Future studies employing more advanced cardiac imaging methods such as Magnetic resonance imaging (MRI) and PET imaging will warrant more accurate and sophisticated cardiac analyses.

3. I disagree with the persistent use of T-test. The authors should consult a statistical advisor. Changes along time such as heart function (Figures 2, 3, 5) should be tested

by 2- repeated-measures ANOVA with appropriate post hoc testing. By using a simple T-test for each time point, the authors artificially increased their chance to find statistical significance. The abuse of simple T-test and wrong statistical analyses are one of the primary reasons for irreproducibility in the present field. The authors should read classical papers in the field in leading journals (Caspi, Huber et al. 2007,) or a recent position paper (Lindsey, Gray et al. 2018).

Response: We thank the Reviewer's comment regarding statistics.

We respectfully claim that we have mainly used one-way ANOVA as a multiple comparison rather than T test in most figures showing heart function (**Figure 2B, 3A, and supplementary figure 6**). We used T test only in **Figure 4-2** which showed the result of TUNEL staining since there were only two experimental groups.

Based on consultation with biostatisticians at City University of Hong Kong and the recommendation from the editor that use of one-way ANOVA instead of 2 repeated-measures ANOVA is appropriate, we have decided to continue using one-way ANOVA as our statistical method.

As the Editor identified, we found some mistakes in describing statistical methods in figure legends, thus we have revised them in the resubmitted manuscript on page 33-36.